



# The sensitivity of intense rainfall to aerosol particle loading - a comparison of bin-resolved microphysics modelling with observations of heavy precipitation from HyMeX IOP7a

Christina Kagkara[1], Wolfram Wobrock[1], Céline Planche[1] and Andrea Flossmann[1]

[1] Université Clermont Auvergne, CNRS, Laboratoire de Météorologie Physique, F-63000 Clermont-Ferrand, France

*Correspondence to*: Wolfram Wobrock (Wolfram.Wobrock@uca.fr)

**Abstract.** Over the Cévennes-Vivarais region in southern France five hour intensive rainfall covering an area of 1000 km$^2$ with more than 50 mm rain accumulation was observed during IOP7a of HyMeX. This study evaluates the performance of a bin resolved cloud model for simulating this heavy precipitation event. The simulation results were compared with

observations of rain accumulation, radar reflectivity, temporal and spatial evolution of precipitation, 5 minutes rain rates and raindrop size distributions (DSD).The different scenarios for aerosol number concentrations range from 1000 to 2900 cm$^{-3}$ and represent realistic conditions for this region. Model results reproduce the heavy precipitation event with respect to maximum rain intensity, surface area covered by intense rain and the duration, as well as the rain DSD. Differences occur in the short-term rain fall rates, as well as for the drop number concentration. The cloud condensation number concentration has

a notable influence on the simulated rainfall, both on the surface amount and intensity but also on the DSD properties and should be taken into account in microphysics parameterizations.

**Keywords.** Intense convective rain, HyMeX, bin microphysics modelling, rain drop size distribution, aerosol particles

## 1 Introduction

Heavy precipitation often occurs in autumn in the Cévennes-Vivarais (CV) region in southern France resulting frequently in

casualties in part due to difficulties of operational weather forecast models to predict their location, timing and amplitude for timely alerts.

In order to evaluate the performance of precipitation forecast model typically radar observations and/or rain gauge measurements are used. Both types of observations were available during the HyMeX campaign (Ducrocq et al., 2014) from September to early November 2012 over the Western part of Mediterranean Sea.

The current study evaluating the model performance will focus on data from IOP7a (Intense Observations Period) observed on 26 September 2012 as it represents the best-documented observational period of intensive rain during HyMeX for the Cévennes-Vivarais region with numerous measurements of precipitation microphysics.



Other modelling studies of heavy precipitation forecast using several intensive observational periods during HyMeX were already performed by Hally et al (2014) for IOP6 and 7a, Duffourg et al. (2016) and Martinet (2017) for IOP16a as well as by Taufour et al (2018) for IOP6 and 16a. A part of these studies focused on the role of dynamical and microphysical processes responsible for heavy precipitation formation, others investigated how the microphysics parameterization impacts

modelled rain forming processes and rain amounts. All studies succeeded in the reproduction of heavy rain events, however sometimes significant differences in location, intensity or microphysical characteristics between model results and observations seem to indicate remaining deficiencies in the physical description of cloud and precipitation formation.

Weather forecast models like IFS/ECMWF use in general bulk parameterizations of the cloud microphysical processes, which treat condensational and collision processes, formation of ice, interaction between the ice and liquid phase but greatly

simplify the dependency of cloud microphysical processes on hydrometeor sizes (Ahlgrimm and Forbes, 2014). More sophisticated bulk parameterizations, as deployed in models like WRF (Thompson et al., 2008; Morrison et al., 2012), COSMO (Seifert, 2008) or Meso-NH (Vié et al., 2016) use two or even three moments schemes wherein the spectra of cloud and rain droplets as well the spectra of ice crystals and precipitating ice particles are prescribed by exponential or gamma distributions. The form of these prescribed spectra can deviate considerably from observed ones although several simulated

bulk parameters, like radar reflectivity, rain and ice water content can sometimes agree with the observations. Studies of Varble et al. (2014) and Taufour et al. (2018) highlight the discrepancies between observed surface raindrop spectra and those simulated with one or two moment microphysics schemes.

Another technique to simulate clouds is the use of so-called bin microphysics schemes. Until today, only few models are available to simulate in a size resolved manner the spectra of drops and ice particles from a few µm to several mm (Geresdi,

1998; Khain et al., 2004; Lynn et al., 2005; Planche et al., 2010) in a 3D meso-scale context due to the often prohibitive computational costs in memory and CPU (Central Processing Units). This detailed methodology, however, allows better insights in the evolution of cloud specific processes such as phase changes and collisional processes and describes more closely the interactions with the field of water vapour and temperature and its feedback with cloud dynamics.

One major objective of this study is, thus, to test if a bin resolved microphysics module in a 3D mesoscale model is more

successful in reproducing a real case of intense precipitation using the dataset obtained during IOP7a of HyMeX than the bulk models that usually simulate intense precipitation, in particular regarding the rain maxima requiring alerts for the population. We focus more specifically on the following questions: is a detailed cloud description (i.e., bin resolved modelling) for all hydrometeor spectra suited to reproduce quantitatively rain accumulations, rain size distribution as well as its spatial and temporal variability that were observed by rain gauges, disdrometer and ground radars?

A further objective is to investigate the influence of aerosol number concentration on surface precipitation and raindrop spectra as it is well known that cloud formation depends on the presence of atmospheric aerosol particles acting as cloud condensation and ice forming nuclei (CCN and INP, respectively). Until today, only few 3D cloud models (see e.g. Leroy et





al., 2009; Thompson and Eidhammer, 2014; Vié et al., 2016) consider aerosol particles in the hydrological cycle, and their model results demonstrated the potential influence of the aerosol concentration on cloud and precipitation development. The bin cloud model *DESCAM* (*Detailed Scavenging Model*, Flossmann and Wobrock, 2010) used in this study follows this approach by forecasting the size distribution of interstitial aerosols and residual aerosols in drops and ice crystals. Aerosol-

cloud modelling, however, needs initial information about cloud condensation nuclei prevailing in the atmosphere prior to the cloud development. These data were available during HyMeX as measurements of the aerosol particle spectra were performed by ground based and airborne observations during the entire experimental period.

The dynamics and microphysics model used in this study, its geographical set-up as well as initial and boundary conditions are described in Sect. 2. In the Sect. 3 the different data used for the comparison with the model outputs from different

observational platforms providing rain parameters and measurements of the prevailing aerosol particle concentration are presented. The current comparisons will focus on the large number of surface observations for the rain while the comparison with airborne data in order to analyse the in-cloud features will be presented in a future work.

The comparison between simulated rain accumulations and results of the Quantitative Precipitation Estimate (QPE) for IOP7a (Boudevillain et al., 2016) is given in Sect. 4. Section 4 also presents the temporal evolution of 5 minutes rain rates

recorded from numerous rain gauge stations, which are analysed, and their evolution is compared to the modelled ones. Finally, in Sect. 5, simulated raindrop spectra are confronted with observed ones from disdrometer measurements. Section 6 summarizes the findings and conclusions of this study.

## 2 Model configuration and model setup

We use for this study the detailed microphysics model DESCAM (*Detailed Scavenging Model*, Flossmann and Wobrock,

2010) which is driven by the 3D dynamics of the anelastic and non-hydrostatic model of Clark et al. (1996) and Clark (2003).

Initial and boundary conditions for horizontal wind, temperature and water vapour mixing ratio were provided by the ECMWF/IFS data products at 00:00 and 12:00 UTC, 26 Sept. 2012. An ensemble study of IOP7a with the mesoscale model Meso-NH (Hally et al., 2014) showed that the application of the ECMWF/IFS data reproduces relatively well onset and

evolution of this precipitation event. The location of the domains for the numerical simulation is depicted in Fig. 1. The outmost model has a horizontal grid resolution of 8 km, for the nested domains the grid size resolution decreases to 2 and 0.5 km, respectively. The vertical grid $z$ is terrain following and non-equidistant. Next to the surface $\Delta z$ is about 40 m and increases to 230 m at 9 km (the outmost model uses a coarser grid, where only every second grid point of the inner vertical grid is used). The outmost domain extends up to 23.5 km; the 2$^{nd}$ and the innermost domain end at 12.5 km. The simulations

in all domains were integrated with a time step of 2 seconds.





The microphysical scheme DESCAM simulates the number distribution functions of aerosol particles, droplets and ice particles as well as the mass distribution of the residual aerosol in drops and in ice crystals. Each distribution function is resolved by 39 classes or size bins, resulting in 195 additional prognostic variables for the numerical model (Leroy et al., 2009). All hydrometeor spectra use a logarithmically equidistant spaced mass coordinate. The resulting drop sizes cover a

diameter range from 2 µm to about 10 mm, aerosol spectra range from 1 nm to 7 µm. For ice crystals the conversion of the mass bins to diameter depends on the chosen mass-diameter relationship (Fontaine et al., 2014). Droplets form when supersaturation occurs, and a subset of the aerosol particles becomes cloud condensation nuclei. After a period of condensational growth larger drop sizes form, initiating collision and coalescence and thus the formation of precipitating drops. Droplets and also aerosols can form ice particles due to heterogeneous and homogeneous nucleation (Meyers et al.,

1992; Koop et al., 2000). The growth of the ice crystals due to deposition of water vapour is treated size dependently and explicitly as a function of the predicted ice supersaturation (i.e., temperature and water vapour mixing ratio). Finally, the collection processes between water and ice (riming) and ice crystals only (aggregation) are also taken into account. For a complete presentation of the microphysical scheme see Flossmann and Wobrock (2010).

In order to provide the model with realistic cloud condensation nuclei we used primarily (*HymRef*) the aerosol number

distributions observed by aircraft measurements on 26 September 2012 (Rose et al., 2015) which took place between 100 and 150 km south of the precipitation event and represents the most polluted conditions encountered during the HyMeX experiment in autumn 2012. The data collected from the different instruments (for details see Sect. 3) were fitted to three log-normal distributions. The parameters for the description of the size distribution (number $N$, mean diameter $D_m$ and standard deviation  ) are given in Table 1. In order to study the role of the prevailing aerosol particle concentration on

precipitation two additional realistic case studies are performed with different aerosol concentrations. The first sensitivity study, called hereafter *HymLow*, uses the aerosol number distribution encountered during a flight on 27 October 2012 (during IOP16 of HyMeX), which also took place over the Northern Mediterranean. IOP16 encountered the lowest particle concentration observed during the entire HyMeX experiment. The minimum and maximum pollution levels observed during autumn over the French Mediterranean coastline thus ranged from 1700 to 2900 particles cm$^{-3}$. As these total numbers are

both quite important a third number distribution with a lower concentration is used. This third size distribution called *Remote*, represents the lowest number concentrations documented by long-term aerosol observations during autumn for the southern part of France from the nearby monitoring station puy de Dôme (Fig. 1; Venzac et al., 2009). The size distribution of 26 September is taken as reference for the following cloud simulations (called hereafter *HymRef*), the other two distributions (*HymLow* and *Remote*) will be used for sensitivity tests.

We know from the aircraft observations that the particle number strongly decreases in the first 3 km of the atmosphere (Kagkara, 2019). This decrease in number concentrations was also represented in the initial conditions for the model simulations. For altitudes above 3 km the aerosol number distribution was kept constant with the values observed at 3 km. The analysis of the aerosol particle composition by mass spectroscopy indicated a slight predominance of insoluble matter.



For the following simulations we assume therefore, that aerosol particles are a mixture of soluble (ammonium sulphate, 40%) and insoluble (silicate-like, 60%) matter of an assumed same molecular weight.

### 3 Observations used for the comparison study

Figure 2 gives a topographical map of the Cévennes-Vivarais region showing the locations of the different ground stations
such as rain gauge, radar and disdrometer locations. We use in this study the quantitative precipitation estimate (QPE) specifically developed for the Cévennes-Vivarais region by Delrieu et al. (2014) and Boudevillain et al. (2016). Radar observations and rain gauge measurements are merged by the geo-statistical technique õ*Kriging with External Drift* (KED)ö providing hourly rainfall data with a spatial resolution of 1 km$^2$ (see e.g. Fig. 3a). The KED precipitation analysis uses 250 hourly rain gauges and four operational weather radars of the French weather service Météo-France. This operational set-up
of the *Cévennes-Vivarais Mediterranean Hydrometeorological Observatory* (OHMCV, Boudevillain et al., 2011) covers an area of 32 000 km$^2$. The region of Fig. 2 only shows S-band radar positions at Nîmes and Bollène and a limited number of rain gauges restricted to the area of interest for precipitation occurring during the IOP7a event. All 31 individual tipping-bucket rain gauges from the õ*service de prévision des Crues du Grand - Delta* ö (SPC-GD, one of the 22 flood forecasting services in France), indicated in Fig. 2 recorded the rain rate with a 5 minutes resolution and will later be used to better
understand the temporal evolution of the intensive rain event.

As DESCAM explicitly simulates the spectra of cloud and rain droplets a comparison of the modelled surface rain spectra with disdrometer measurements is also attempted. Two disdrometers, located at La Souche and Saint-Etienne-de-Fontbellon (StEF) (see Fig. 2) encountered strong precipitation during IOP7a. All rain spectra were measured by OTT Parsivel2 disdrometers with a time resolution of one minute. The observed size distributions described in Sect. 5. were corrected
following the method proposed by Raupach and Berne (2015).

Observations with a fast scanning small range X-band radar were also included in our comparison as they provide spatially and temporally high resolved reflectivity fields (60 m in radial direction, 30 s for one PPI (Plan Position Indicator)) and thus give insights in the small-scale dynamics of the precipitating system. The X-band observations used for this study stem from the X3 radar at La Bombine located at the northwest rim of the Cévennes (Fig. 2) at an altitude of 975 m. Its PPI coverage is
restricted to a single elevation of 1.5° and its horizontal range is 36 km (see Fig. 4).

The number distributions of aerosol particles were measured during numerous IOPs of HyMeX by the French research aircraft ATR-42 (Rose et al., 2015). The airplane was equipped with an instrumental set-up including a Scanning Mobility Particle Sizer (SMPS) and a GRIMM Optical Particle Counter (OPC). The SMPS provides particle size distributions with diameters from 20-485 nm in time intervals of 130 s, whereas the GRIMM OPC detects particles in the size range from 300
nm to 2 m every 6 s. All the size distributions were recorded at altitudes between 200 and 3700 m. For the simulations





presented in this study we use observations made in the afternoon of 26 September 2012, about 150 km south of the foothills of the Cévennes-Vivarais just over the northern range of the Golfe du Lion (see Fig. 1).

## 4 Results

IOP7a was the most intense rainfall episode observed during HyMeX in autumn 2012 over the Cévennes-Vivarais region
(Figs. 1 and 2). A low-pressure system over the British Isles favoured the development of a south-westerly flow of warm and humid air which persisted in the free troposphere up to an altitude of 8 km. In layers above 8 km the flow became more westerly. The 0°C isotherm level was situated at an altitude of 3.7 km. In the lowest atmospheric layers (below 900 hPa) airflow from south and south-eastern directions converged with the westerly flow over the coastal area of the Golfe du Lion. Consequently, convective cells developed over the south-eastern rim of the Massif Central (i.e. the relief of the Cévennes-
Vivarais). First episodes of heavy rain started shortly after 6:00 UTC almost simultaneously for the rain gauges 8, 12, 21, 26 and 29. Especially these two northern stations 26 and 29 encountered heavy rain continuously until 10:30 UTC. A second intense convective period took place from 7:30 to 9:30 UTC in the area of rain gauges 9, 10, 13-15, 19 and 23 (i.e. close to the X-band radar observational area), where rainfall varied locally for these places between 25 to 70 mm and lasted more than 2 hours. (Details about the time evolution of local precipitation will be illustrated in Sect. 4.4).

Convective rain fall occurred until 11:00 UTC and terminated due to the arrival of a cold front which brought stratiform and less intense precipitation in the afternoon over the Cévennes-Vivarais region. For a more detailed description of the meteorological conditions see Ducrocq et al. (2014).

Figures 4a-c show simulated and observed X-band radar reflectivity fields obtained for a beam elevation of 1.5°. Radar reflectivity in Fig. 4a is calculated from the results of simulation *HymRef* for the innermost domain, using the sixth
momentum of the modelled hydrometeor size distribution to determine the normalised radar reflectivity $Z_{dBZ}$ (Planche et al., 2010). Here $Z_{dBZ}$ at 7:50 UTC was selected, as it reflects well the predominant orientation and the spatial distribution of the precipitating cells over the northern Cévennes and the Vivarais.

The X-band radar observations only cover the south-western part of the innermost model domain and allow the comparison for a quite limited area of precipitation (marked by the circular surface of 60 km in diameter) encountered during IOP7a.
Reflectivity observed at 7:50 UTC (Fig. 4b) shows that precipitating cells have the same orientation as in the model. Compared to the measurements the modelled rain band inside the radar range is slightly shifted to the east and the convective cells south from the radar position are less important than the observed one. For the simulation *HymRef* the formation of the convective cells starts (at 7:20 UTC) in this southern range of the radar but rainfall is still weak. Rain gauge observations however indicate that strong convective showers already occurred 20 to 30 km south of the radar location.





Figure 4c shows the convective cells that develop around 9:00 UTC during the second convective period. Strong convection formed at this time further to the north, next to the radar location, and cells propagate in northward direction. This deviation from the dominantly south-western track of rain cells could not be reproduced in the model.

In the following sections we will compare the model results with the surface observations of rain accumulation, rain rates and droplet spectra and address also the differences in the model results for the three aerosol scenarios.

### 4.1 Rain accumulation

Figures 3a-d show rain accumulation for the period from 6:00 to 11:00 UTC as determined by the KED analysis and simulated by the cloud model for three different scenarios of aerosol particle concentrations. Figure 3, in contrast to Figs. 1 and 2, uses the kilometric coordinates of the third model domain (i.e. the innermost domain). Precipitation after 11:00 UTC was still ongoing but appeared only very locally and with low intensity ($< 2$ mm h$^{-1}$). The KED results indicate that largest rain accumulations with 115 mm occurred over the Vivarais Mountains. A small second maximum with 76 mm developed 20 km to the north. The location of the main maximum is reasonably reproduced by all simulations (Figs. 3b-d). Spatial deviations between observed and modelled rain maxima are within a radius of 5 to 7 km. The orientation of the modelled rain band is quite similar to the observed one. The location of the secondary maximum in the northern precipitation field is also reproduced especially for the simulations with the intermediate aerosol loading case (*HymLow,* Fig. 3d).

Strongest difference between observations and model results occur in the surface extension of rain. We note that the width of the observed rain band is much larger than the simulated one as it extends more to the west. This area of precipitation with a local maximum of 74 mm of rain at $(x,y) = (540,640$ km$)$ was caused by the second convective phase from 9 to10:30 which was less pronounced in all simulations.

The differences in surface rain extension between observation and model becomes most obvious when the total water mass of the 5 h rain accumulation (i.e. from 6:00 to 11:00 UTC) is calculated by integration over the entire domain displayed in Fig. 3a. Table 2 gives the surface integrated total mass of rain water resulting from the KED analysis and from the simulations *HymRef*, *HymLow* and *Remote* accumulated until 11:00 UTC. In the Table 2, different thresholds of rain accumulation are used to calculate the precipitation amounts. When considering precipitation events with a threshold exceeding 2, 10 or 20 mm for each model grid point, we note that the simulated rain accumulation remains 40 to 50 % smaller than the observed ones. Restricting however to the areas with only strong precipitation where rain accumulation exceeds 50 mm, differences with the simulations are reduced to 10 to 20% only. This result can also be expressed in terms of the area coverage of precipitation: while the observed surface accumulating of at least 10 mm covers 6300 km$^2$ accordingly to Fig. 4a, the model results only in 2100 km$^2$ for *HymRef* and in 2700 km$^2$ for *Remote*. Surface accumulating of more than 50 mm, however, were observed for an area of 1060 km$^2$, while the simulated areas are close with 700 to 820 km$^2$ depending on the scenario (*HymRef* and *Remote*, respectively).





### 4.2 Spatial and time evolution of the rain field

The QPE, determined by means of the KED technique, provides the rain accumulation on an hourly basis. Figure 5a illustrates the hourly rain accumulation between 7:00 and 11:00 UTC. Four surfaces coloured in red, green, orange and blue display the temporal shift of the rain field from the east to the west in increments of 1 hour. The coloured surfaces delimit

regions with rain accumulation larger than 20 mm per hour.

Figures 5b and c show the same results for the simulation with the strongest (*HymRef*) and the weakest aerosol loading (*Remote*). We can see that the shift of the rain field from 7:00 to 11:00 UTC is less pronounced than in the KED analysis of Fig. 5a. This is again a consequence of the underestimated second convection zone to the west and to the south as already indicated in Sect. 4.1. Furthermore, it becomes evident that regions with maximum rain accumulation are those where rain

lasted for more than 2 or 3 hours. Figure 5c shows that the rain pattern is mostly steady-state in the *Remote* case which finally results in the strongest rain accumulation as already noted in Table 2. At the end of the three different scenarios, as described in the Sect. 4.1, the spatial rain accumulation field shows some differences, especially in the location of their maximum. The Taylor diagram (Fig. 6) visualizes the skills of the different scenarios in simulating the rain accumulation field. The *HymRef* scenario simulates the spatial pattern of precipitation quite well and better than the simulations using the

two other aerosol particles loadings.

### 4.3 The effect of the initial aerosol particle number distribution

Comparing the results for the different simulations in Table 2, it becomes obvious that the total rain mass is highest for the *Remote* aerosol particles distribution, then decreases for the case *HymLow* where aerosol particle number increased by a factor of two and is the lowest for the highest aerosol number concentration of *HymRef*. The last line in Table 2 gives the

observed and simulated maxima of rain accumulation. The maximum of rain accumulation for *Remote*, i.e. cleanest atmospheric conditions, exceeds the results for *HymRef* and *HymLow* considerably.

The increase in rain accumulation with decreasing initial aerosol concentration is associated with an increase in strength and intensity of the main rain field to the south due to an earlier onset of rain and an intensification of the rain amount in zones with weak precipitation (see e.g. isolated rain area in the south-eastern corner of the model domain in Fig. 3).

The differences in rain accumulation for the three aerosol scenarios result from the activation and condensation processes in the initial phase of cloud development, which is driven by the number concentration of cloud condensation nuclei. Under atmospheric conditions with low particle concentrations droplets can form precipitation more rapidly as the field of water vapour supersaturation becomes stronger (Planche et al., 2010). Consequently, also the development of the ice microphysical processes, as well as the latent heat release, are modified causing changes in the dynamical development of the simulated

cloud.





Detailed studies of the effect of aerosol particle number and solubility on precipitation formation have already be done with the same dynamical-microphysical model DESCAM for individual short living convective systems over Central Europe (Planche et al., 2010) as well as over Florida (Leroy et al., 2009). In these previous studies the rain duration was much shorter (< 30 min) and accumulation did not exceed 30 mm. The results of the present study for orographically generated and

long-lasting steady state convection confirm our previous findings: more rain occurs when low particle numbers prevail.

Figure 7 displays the spatial frequency distribution (PDF) of observed and simulated 5 h integrated rain accumulation in the range from 10 and 150 mm. The comparison of this distribution function displays a considerable similarity between simulations and KED analysis, i.e., a strong decreasing trend of the frequency with increasing rain intensity. A closer look to Fig. 7 confirms the previous results that the underestimation of simulated rain mainly occurs in the range of weak to medium

accumulation, i.e. from 15 to 40 mm (note the logarithmic axis of Fig. 7). In addition, we can detect that the frequency of rain events > 70 mm is for all model results higher than in the KED analysis.

The differences in the frequency distribution between the three aerosol cases *Remote*, *HymLow* and *HymRef* confirm the role of the particles number concentration on the amount of the rain accumulation. Using the number concentration for *Remote* conditions, a rain accumulation of up to 154 mm is simulated by the model. The number of grid points with more than 120

mm remains however quite low (0.1% of the simulated rain surface). For the two simulations with the higher particle concentrations observed during HyMeX (*HymLow* and *HymRef*) the maximum rain accumulation does not exceed 120 mm. However, the rainfall from the lower aerosol concentration *HymLow* exceeds the total rain mass from the *HymRef* case (see Table 2). Figure 7, thus, confirms that the increase in total precipitation is caused by an increasing number of locations with high rain accumulation (larger than 60 mm).

**4.4 Time evolution of local precipitation**

In a further step we evaluate the capacity of the model to reproduce the temporal variability of the rain. Herefore, we use the rain gauge measurements whose positions are indicated in Fig. 2 by the numbers 1 to 31. They recorded temporally highly resolved rain rates during 5 minutes interval. Measurements for four stations with long lasting and intense rain periods are displayed in Fig. 8. These stations are all located in the Vivarais region where strongest rainfall occurred during this day

(compare Fig. 3a).

Rainfall started shortly after 6:00 UTC over the most northern stations St. Félicien (29) and Lamastre (26) (Figs. 8a and b). The first precipitation period mainly concerned the eastern part of the Vivarais. The rain gauge of Le Cheylard (24) (Fig. 8c) collected during this initial period only low quantities of rain. At 7:30 UTC, however, another convective rain cluster formed slightly more to the west, extending from gauge 13 over 19 to gauge 24. Figures 8c and d illustrate this second important rain

period for gauges 19 and 24. Radar observations for this zone reported persistent and strongest reflectivity resulting in the maximum rainfall in the KED analysis. Unfortunately, no other observational sites are located between the gauges 17, 19, 21 and 24 confirming this maximum.



Figure 9 displays the time series of simulated 5 min rain rates for the rain gauge locations 19 and 24. Results for the *HymRef* scenario are given in Figs. 9a and b, those for *Remote* in Figs. 9c and d. Rain accumulation for the southern location 19 gave 51 mm in the *Remote* simulation and 47 mm in *HymRef*. Differences between both scenarios are more pronounced for the northern location 24 where 112 mm were simulated for the *Remote* case but only 90 mm for *HymRef*. The temporal evolution of other grid points can of course deviate from these, but the ones selected for Fig. 9 document quite well the overall characteristics of the modelling results.

We note that when comparing the modelled time series with the observed ones two features are most striking:

(1) the amplitude of the observed rain rates are more fluctuating than the simulated ones, that means, the simulated local rain evolution shows a more continuous increase and decrease and,

(2) the maxima of the observed 5 min rain rate attain higher values than in the simulations.

We detect the absence of rain rates exceeding 8 mm (Figs. 8a and d) as the modelled rain rates (Fig. 9) generally stay below 4-5 mm per 5 min. For this comparison, we need to consider that the model results represent rainfall over a grid box of 500 x 500 $m^2$ while rain gauges have a collection surface only of about 0.04 $m^2$.

In the observations, as well as the model, however, the appearance of very strong rain events (with more than 7 mm/5 min) is always preceded by 15 to 20 minutes of rain of moderate intensity. The time evolutions as presented in Fig. 9 confirm that the simulated areas with strong precipitation are caused by long lasting and continuous rain episodes, while the observations indicate stronger intensities over shorter time intervals, a feature which is sub-grid in the model and, thus, cannot be confirmed.

Comparing the results of the *HymRef* with the *Remote* scenario it becomes evident that not only the rain accumulation but also the strength of the 5 min rain rates increased in the presence of less cloud condensation nuclei. Maximum rain rates with more than 8 mm per 5 minutes were recorded several times by the rain gauges. Model simulations for high aerosol numbers of *HymRef* do not exceed rain rates larger than 6 mm, while the simulations with the low aerosol numbers (*Remote*) can reach up to 9 mm. A frequency analysis of the 5 min rain rates (Fig. 10) summarizes the differences between the model scenarios and the observations. This figure confirms the presence of rain rates stronger than 6 mm/5min in the observations as well as its absence in the simulation *HymRef* with high aerosol number concentrations. The model scenario *Remote* can produce the rain rates larger than 6 mm but their occurrence remains significantly below the observed ones.

In addition, the frequency analysis in Fig. 10 also demonstrates that the model produced more often rain in the range of 1.5 to 4.5 mm/5min, which is finally responsible for the strong rain accumulation obtained in the simulations. The different scenarios lead to modified developments of the precipitation fields. In addition to a rain rate increase, we note a modification in the surface area covered by rain during the intense convective period (see Figs. 3 and 5).





## 5 Comparisons at raindrop scale

As DESCAM is a bin resolved cloud model, we attempt in a final step of our model evaluation a comparison between simulated and observed raindrop size distributions (DSD). During IOP7a two disdrometers counted raindrops, one at La Souche in 950 m and the other at StEF at 350 m asl (see Fig. 2). The disdrometer at StEF recorded rain spectra from 6:30 to

7:30 UTC, the one at La Souche from 7:30 to 9:10 UTC, both with a 1-minute time resolution.

The shape of the number distribution varied essentially with total rainwater content (RWC) whose values reached up to 7 g m$^{-3}$ when integrating over the observed DSD. We restrict however our analysis to the spectra below 3.5 g m$^{-3}$ as such spectra already provide high rain rates from 6 to 11 mm/5min depending on the size of the mean mass diameter.

Figure 11a shows the DSD of the La Souche disdrometer measurements distinguished into four categories of RWC: 0.5, 1, 2

and 2.9 g m$^{-3}$. The spectra displayed are averages over all DSD for which RWC deviates ±20% from the selected mean value of 0.5, 1 and 2 g m$^{-3}$. However, the DSD of RWC of 2.9 g m$^{-3}$ represents a mean for spectra holding 2.7 to 3.5 g m$^{-3}$. Figure 11a illustrates that the increase in RWC is accompanied by the presence of larger drops. In addition, we can see for the spectra with 2 and 2.9 g m$^{-3}$ the number of droplets < 1 mm is significantly higher than for the spectra with low RWC. Figure 11c displays the corresponding mass distributions of the DSD given in Fig. 11a. In order to better illustrate the mass

contribution of the different drop sizes to the RWC, we plotted both axis of the mass spectra in a linear scale. At first glance we can discover that drops smaller than 1 mm only contribute very little to the averaged RWC and the mean mass diameter shifts from 1.3 mm for the observed DSD with 0.5 g m$^{-3}$ to almost 3 mm in the case with strong RWC of 2.9 g m$^{-3}$.

Figure 11b and d show corresponding illustrations from the model simulation. The model data used for this comparison occurred between 7:50 to 10:00 UTC. We considered simulated DSD only from surface grid points where the topographical

elevation was between 900 and 1000 m and modelled RWC ranged around 0.5, 1 and 2 g m$^{-3}$ (also within an interval of ±20%). The results restrict to the DSD simulated by the scenario *HymRef*. The maximum RWC modelled in this case at this elevation never exceeded 2.4 g m$^{-3}$ which explains the absence of the 2.9-curve (in the scenario *Remote*, which is not presented here, stronger rain events with 2.7 g m$^{-3}$ between 9000 and 1000 m were encountered).

The simulations of number and mass distribution clearly demonstrate that the sizes of the simulated raindrops increase with

increasing RWC. In contrast to the observations the number concentration for the smaller drop sizes, however, decreases with increasing RWC. This behaviour in the model is associated to the fact that large raindrops can only form through collision-coalescence with smaller precipitating drop sizes. In the simulation, cloud base is located between 1300 to 1400 m asl, i.e. about 400 m above the ground, and thus no cloud droplets are present at the surface. Figure 11b illustrates quite well this process as the larger sizes increase at the cost of smaller raindrops in the range from 0.3 to 1.5 mm. Concerning the

observations, we need to keep in mind that the location of La Souche is in a mountainous region of 950 m. We cannot exclude that the observational site was closer to cloud base or even immersed in the cloud, explaining the presence of an increased number of small raindrops. A comparison with the raindrop spectra observed at StEF in 350 m asl confirms


partially this hypothesis as the number concentration for drops < 1 mm is generally a factor of 2.3 smaller (Zwiebel et al., 2016), a decrease of this number with increasing RWC, however, could also not be detected.

The comparison between observed and modelled DSD was restricted so far to the scenario *HymRef* presenting a relatively high concentration in aerosol particles. In Fig. 12 we compare the mass distributions resulting from all three different

scenarios already discussed above. In order to get a statistically reliable result we only compare data at surface level ranging between 500 and 600 m as rain occurred most frequently in this elevation in the time span from 7:30 to 10:00 UTC. As a consequence, each mass spectrum is an average over 2800 to 3200 individual DSD. Their mean RWC is 0.88 g m$^{-3}$ for *HymRef* as well as for *HymLow*, but 0.96 g m$^{-3}$ for the *Remote* scenario. We note from Fig. 12 that the initial aerosol number concentration also influences the final DSD. The mean mass diameter shifts from 2.2 mm for *HymRef* to 2.9 mm for the

lowest initial aerosol distribution (*Remote*). The simulation with about 1700 particles cm$^{-3}$ is located between the two other cases.

## 6 Summary and Conclusion

A major objective of this study was to test if a bin resolved microphysics module in a 3D mesoscale model is successful in reproducing a real case of intense precipitation. The heavy precipitation event in the Cévennes-Vivarais region observed

during the HyMeX field experiment IOP7a in autumn 2012 was selected and analysed. Results for the QPE indicate a maximum rain accumulation of 115 mm during 5 h over the Vivarais Mountains. Heavy precipitation with more than 50 mm covered a surface of about 1060 km$^2$. The high quantity of rain was caused by the permanent formation of new convective cells over the same mountainous barrier.

The simulation with the bin-resolved cloud model produces quite well the location of the rain maximum and also the surface

area covered by heavy precipitation (> 50 mm). In the surrounding regions with lower rain amounts, however, model results underestimate the superficial rain area by more than a factor of 2. A comparison of the temporal development of the observed rain field shows that the triggering of convection occurred for a wider spatial spread than in the simulation.

We suspect that the initial and boundary conditions imposed by the IFS/ECMWF data at 0:00 and 12:00 UTC with a grid resolution of 0.25° provided a too homogenous structure for the fields of wind, temperature and humidity. During the

integration time from 0:00 to 6:00 UTC prior to convection formation the model could not produce the small mesoscale atmospheric heterogeneities which exist under real conditions. These differences, e.g. in atmospheric humidity, become obvious in the temporal and spatial delayed onset of rain, as its formation starts almost one hour before and 25 km more to the south than in the simulations (Fig.3).

Model simulations were initialised with three different scenarios of aerosol number concentrations. For the reference case

(*HymRef* with 2900 cm$^{-3}$) aircraft observations of aerosols sampled 15 h prior to the rain events and about 150 km upwind of the convection formation were used. The second scenario prescribed the lowest number concentration of *HyMeX* in autumn





2012 with 1700 cm$^{-3}$ (*HymLow*) sampled by aircraft measurements, and the third one (*Remote*) the lowest concentration of 1000 cm$^{-3}$ available from long-term observations collected in the North of the French Massif Central. It was found that the decrease in the aerosol concentration from 2900 to 1000 cm$^{-3}$ enhances rain accumulation by 20 % and also the area covered by heavy precipitation (see Figs. 3 and 5). In addition, a frequency analysis of the spatial distribution of the rain

accumulation shows that the gain in precipitation for low aerosol loadings is mainly caused by the increase of the number of locations with rain accumulations > 60 mm.

In a further step of our analyses we compared the local behaviour of the modelled precipitation with measurements of 5 minutes rain rates sampled from 31 individual gauges. The temporal evolution of observations indicates that the highest rain rates of 5 to 9.5 mm/5min do not appear immediately but only after a period of weak to medium rainfall lasting at least 15 to

20 minutes. Model results presented in Sect. 4.4 confirm this observational result.

After achieving their maximum the observed rain intensities drop significantly between 5-9 mm in the following 5 minutes and thereafter rain ceases or remains low until new convective precipitating cells occur. This abrupt drop in rain rate could not be reproduced in the model. Decreasing intensities of 6 mm/5min can also occur in the simulations but rainfall continues generally without intermittency. Consequently, the simulated rain rates are less fluctuating and more continuous than the

observed ones, also because they represent a mean value over a 500 m grid, in contrast to the more localised (i.e. punctual rain gauge) observations.

Comparisons with spatially high resolved X-band radar observations indicate that the modelled grid resolution is not sufficient to resolve the fine scale structure of the convective cells encountered. Consequently, the modelled convective clouds are less fluctuating, and the resulting rain is of more continuous character and maximum rain rates are probably less

strong than the observed ones.

The analysis of the rain rate for the different aerosol scenarios also highlights differences occurring when low or high number concentrations were used. Rain intensities reach up to 9 mm/5min for simulations of the *Remote* case while only 6 mm/5min were simulated when high particle concentrations were used for the *HymRef* scenario.

One of the main weaknesses in cloud microphysics modelling is generally the simulation of the raindrop size distribution

(DSD) as most cloud models with parameterized microphysics have significant problems to realistically reproduce shape, number and mass of the rain DSD (Varble et al., 2014; Taufour et al., 2018; Tridon et al., 2019; Planche et al., 2019). The shape of the number and mass size distributions presented in Sect. 5 compare quite well with disdrometer measurements sampled for this precipitation event. The observations illustrate as expected that the rain DSD becomes wider with increasing rainwater content (RWC). This is also well simulated by the bin resolved modelling. The analysis of the disdrometer

measurements shows in addition that the number concentration of all droplet sizes (starting at diameters of 0.3 mm) increases with increasing RWC. This behaviour could not be detected in the simulated spectra as the number concentration for drop diameters smaller 1.5 mm decreases with increasing RWC. In order to explain this model results we note that the





simulated cloud base was located about 400 m above the ground. Due to the cut off from smaller raindrops prevailing inside the cloud the evolution of the DSD was mainly determined by collision-coalescence, which increases the size of the large drops and reduces the number of the small ones.

DSD measurements presented in Sect. 5 are located at 950 m asl. We cannot rule out that cloud base was sometimes below

this elevation and the instrument was immersed into the cloud at least part of the time where numerous smaller drops are present. The analysis of a second disdrometer, located at 350 m asl, supports this hypothesis. For RWC of 2 g m$^{-3}$ number concentration for drop sizes from 0.3 to 1 mm ranged typically between 700 to 1000 m$^{-3}$ and thus remains significantly below those counted for the mountain station La Souche in 950 m. But even for this second disdrometer an important reduction in small raindrops as simulated by the model cannot be reproduced. Also, other observational studies on strong

precipitations events (Raupach et al., 2019 or Thurai and Bringi, 2018) confirm the behaviour that drop size concentration typically does not decrease with decreasing drop diameter.

A possible explanation of this difference in the modelled rain spectra could be attributed to the treatment of breakup process of large raindrop sizes in the model. At the moment only spontaneous breakup is treated accordingly to Hall (1980) and becomes active for droplet sizes larger than 5 mm. We cannot exclude that in particular under the conditions of strong RWC

also collisional breakup is occurring (Low and List, 1982) and the redistribution of the õbrokenö drops will privilege the formation of smaller drop sizes. Another possible explanation for the underestimation of the small raindrop sizes can be caused by the lack of corresponding cloud particles in elevated cloud layers. A comparison of the modelled ice particle distribution in altitudes around -12°C (or 5 km) with aircraft observations indicates an underestimation of the number of ice crystals smaller than 1 mm. A detailed comparison with airborne data in order to analyse the in-cloud features, in particular

related to the ice phase, will be presented in a future work.

Finally, also for the raindrop size distribution the influence of the aerosol loading can be detected. As presented in Sect. 5 we can see that the mean mass diameter increases from 2.2 mm for the strong aerosol concentration of *HymRef* up to 2.9 mm for the lowest aerosol charge in the *Remote* scenario. Thus, the variation of 2900 to 1000 aerosol particles per cm$^3$ results in a significant modification of the mean mass diameter of the rain DSD. Information regarding the cloud condensation nuclei

number should, thus, also be taken into account in parameterized models.

However, the differences encountered for the modelling of rain accumulation, rain rate as well as raindrop spectra remain quite small when restricting our comparison to the aerosol concentration (i.e. *HymRef* and *HymLow*) that really were encountered during HyMeX. Only a further important reduction in the particle concentration to remote continental conditions highlights the potential role of the aerosol particle number.

Regarding the other objective of the current investigation, our study showed the potential of a bin-resolved modelling to reproduce the heavy precipitation periods. Even though the weaker precipitation was underestimated in the model, the peak values that would warrant an alert to the population were better represented as they are generally in bulk models.



In order to improve the bulk models for routine forecast, the microphysical parameterizations should probably include a dependency on the cloud condensation nuclei distribution, as well as a possible evolution of the form of the hydrometeor spectra. Bin resolved models like DESCAM could provide some guidance for their development.

*Acknowledgement.* This work is a contribution to the HyMeX program and MUSIC project, supported by Grants ANR-14CE1-0014 and MISTRALS/Hymex. C.K was funded by the MUSIC project. The model calculations have been done on French computer facilities of the Institut du Développement des ressources en Informatique Scientifique (IDRIS) CNRS at Orsay, the Centre Informatique National de l'Enseignement Supérieur (CINES) at Montpellier under the project 940180 and

10    the Centre Régional de Ressources Informatiques (CRRI) at Clermont-Ferrand. The authors acknowledge the HyMeX database teams (ESPRI/IPSL and SEDOO/OMP) for their help in accessing the data. The authors extend their gratitude especially to Brice Boudevillain and the OHMCV observatory for providing the KED data, as well as to Evelyn Freney for the help regarding the information about initial aerosol population.



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





**Table 1.** Aerosol particle number distributions used for the different modelling scenarios. Each size distribution is composed of three modes (*i*) defined with a log-normal distribution using the number $N$, the mean diameter $D_m$ and the standard deviation parameters ($N$ in cm$^{-3}$ and $D_m$ in µm).

| mode $i$ | *HymRef* (26 Sept. 2012, IOP7a) | | | *HymLow* (27 Oct. 2012, IOP16) | | | *Remote* (long term obs., puy de Dôme) | | |
|---|---|---|---|---|---|---|---|---|---|
| | $N_i$ | $D_{m,i}$ | $log\ \sigma_i$ | $N_i$ | $D_{m,i}$ | $log\ \sigma_i$ | $N_i$ | $D_{m,i}$ | $log\ \sigma_i$ |
| 1 | 2900 | 0.06 | 0.26 | 1400 | 0.05 | 0.26 | 150 | 0.025 | 0.146 |
| 2 | 72 | 0.32 | 0.20 | 300 | 0.16 | 0.198 | 610 | 0.52 | 0.217 |
| 3 | 3 | 0.72 | 0.397 | 4 | 0.72 | 0.396 | 250 | 1.35 | 0.176 |





**Table 2.** Total mass of rainwater (in Mtons) recorded and simulated for the period from 6:00 to 11:00 UTC, 26 Sept. 2012 for different thresholds of rain accumulation. The bottom line gives the local maximum values occurring for observed and simulated rain accumulation (in mm).

| Rain amount | Observation (KED analysis) (in Mtons) | HymRef (in Mtons) | HymLow (in Mtons) | Remote (in Mtons) |
|---|---|---|---|---|
| > 2mm | 209.9 | 102.3 | 117.0 | 121.4 |
| > 10 mm | 190.3 | 83.9 | 97.8 | 101.7 |
| > 20 mm | 149.0 | 73.7 | 86.1 | 87.7 |
| > 50 mm | 69.7 | 52.4 | 58.9 | 63.2 |
| max (in mm) | 116 | 118 | 117 | 150 |

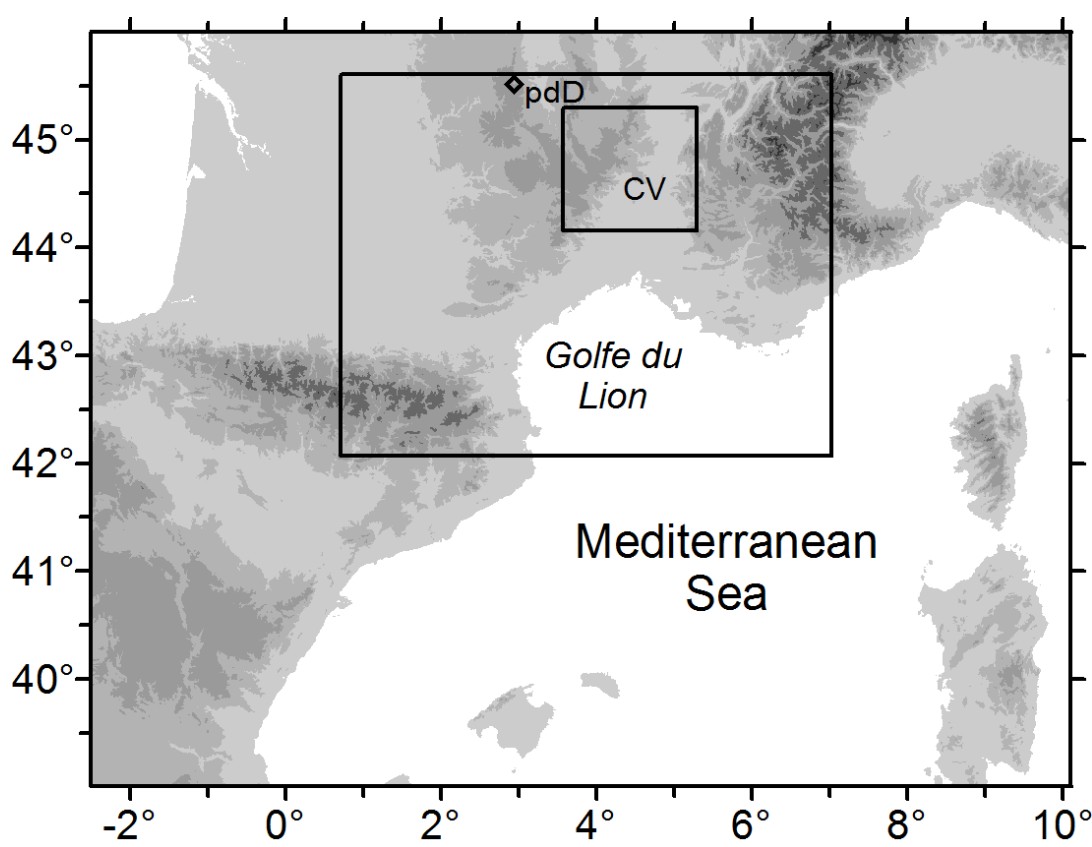

**Figure 1.** Map of the three model domains. CV corresponds to the region of the Cévennes and the Vivarais and pdD to the

5   Global Atmosphere Watch station on the puy de Dôme.
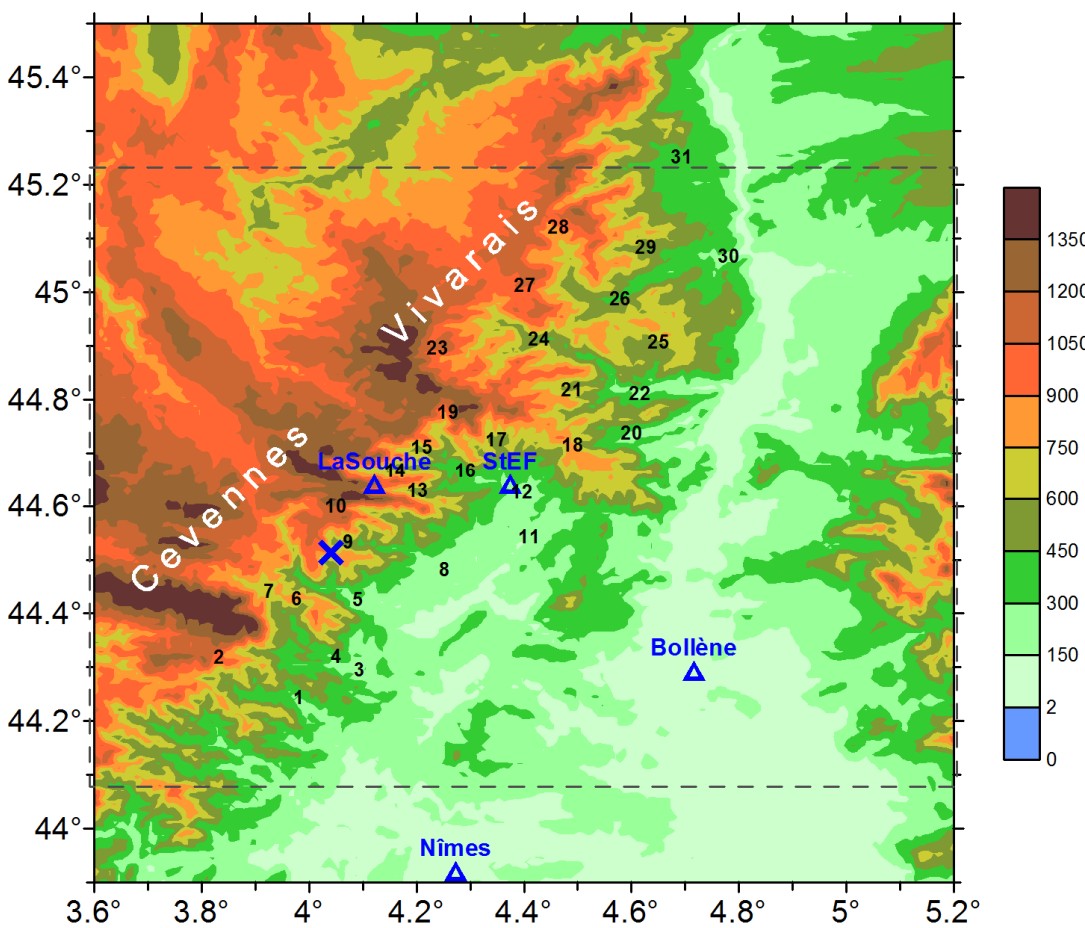

**Figure 2.** Topographical map of the region (altitude in meters) of strong precipitation during IOP7a. Numbers indicate the locations of rain gauges used for this study. The dotted lines show the innermost model domain. Disdrometer measurements were done at La Souche and at Saint-Etienne-de-Fontbellon (StEF), S-band radars of Météo-France are located at Nîmes and Bollène, and X-band radar is located at La Bombine indicated by the blue **X**.


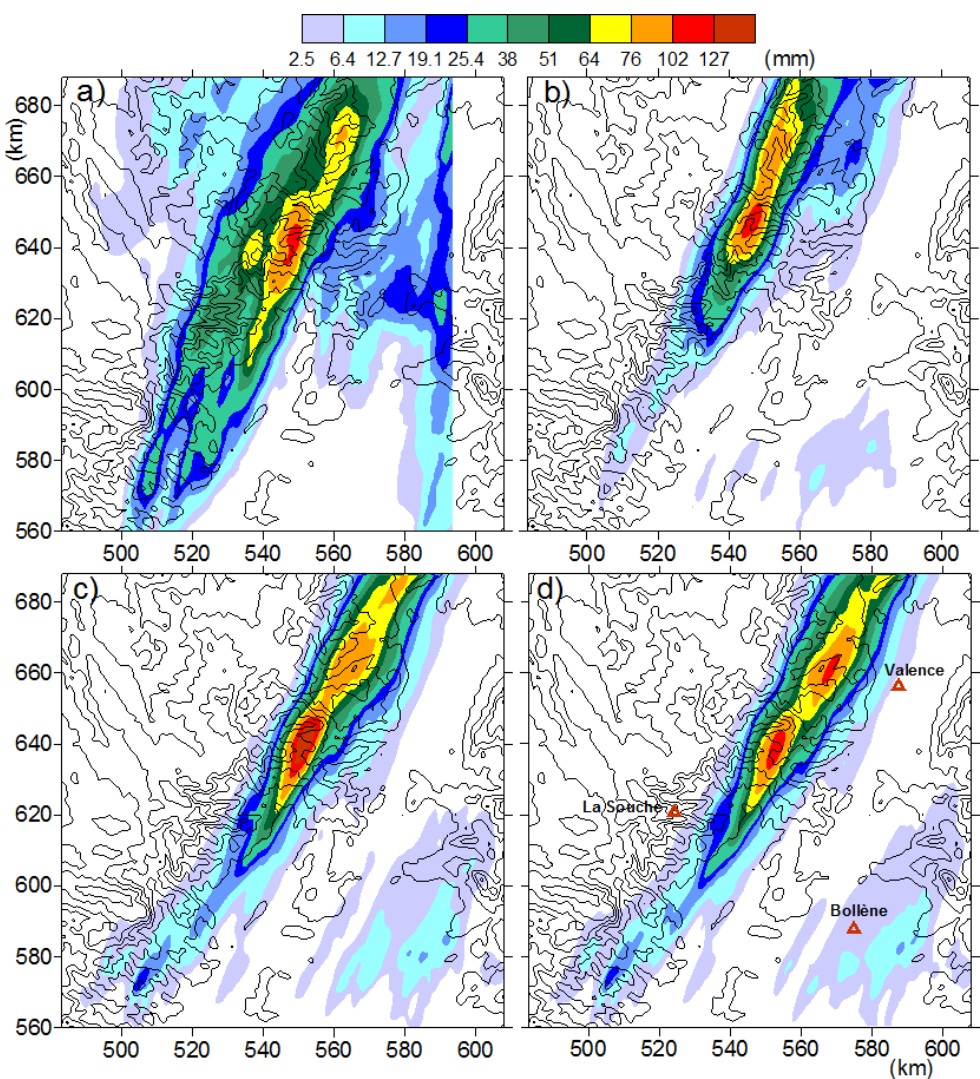

**Figure 3.** Observed and simulated rain accumulation for 26 Sept. 2012, at 11:00 UTC. a) corresponds to the observations
5 (i.e. KED analysis), b) to the simulation using *HymRef*, c) to the simulation *Remote* and d) to the simulation *HymLow*.

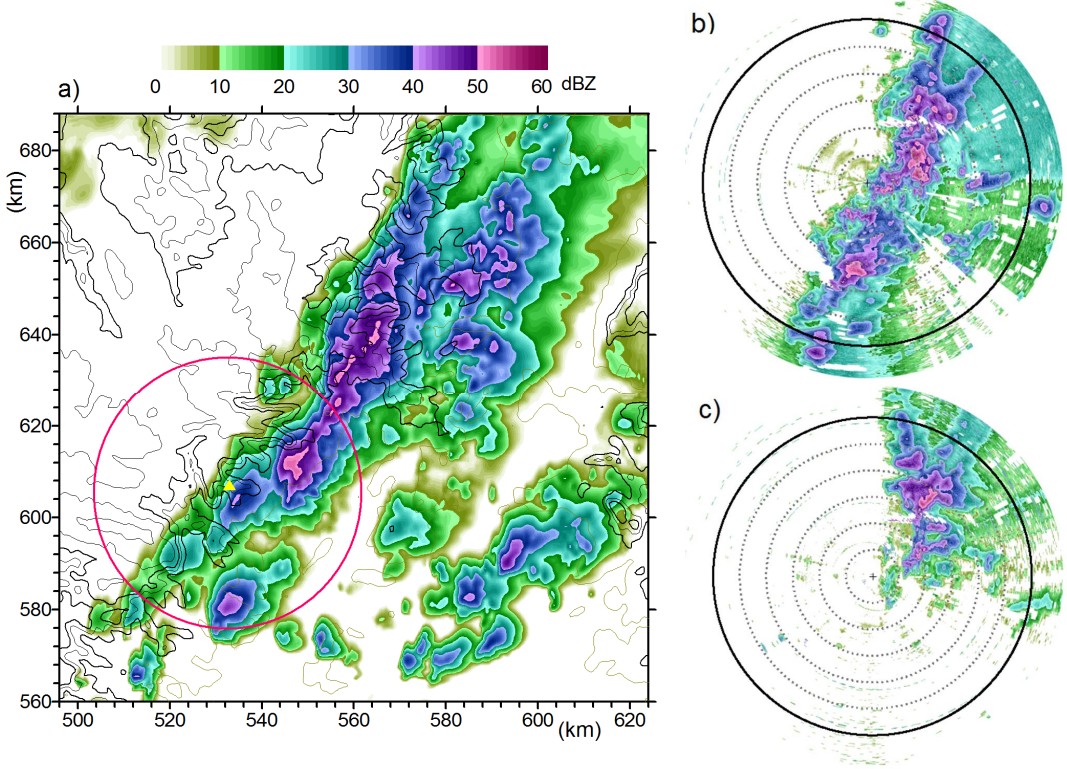

**Figure 4.** X-band radar reflectivity simulated for the *HymRef* scenario at 7:50 UTC (a) and observed by the X3 radar at 7:50 UTC (b) and at 9:06 UTC (c). The red circle in a) displays the 30 km radius range of the X-band observations.

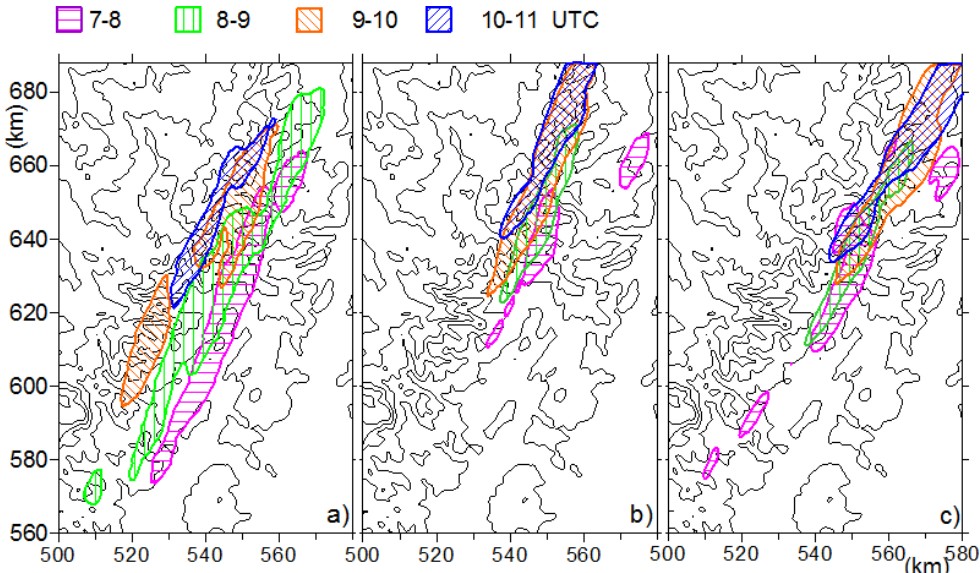

**Figure 5.** Surface areas with hourly rain > 20 mm. Results from the KED analysis are presented on a), the model results for

5   scenario *HymRef* in b), and those for scenario *Remote* in c).

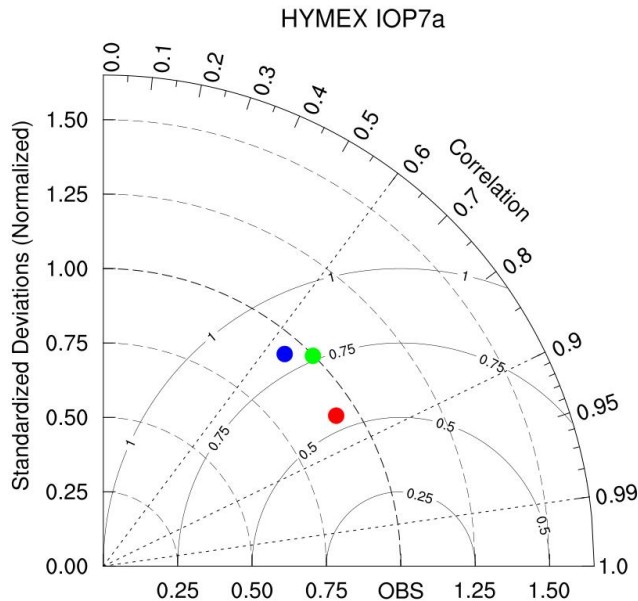

**Figure 6.** Taylor diagram for the rain accumulation from the three different scenarios: *HymRef* (in red), *HymLow* (in blue) and *Remote* (in green). The radial coordinate shows the standard deviation of the rain field, normalized by the observed standard deviation. The azimuthal variable shows the correlation of the modelled rain accumulation field with the observed one. The distance between the reference KED data (i.e. OBS) and individual points corresponds to the root mean square error (RMSE).




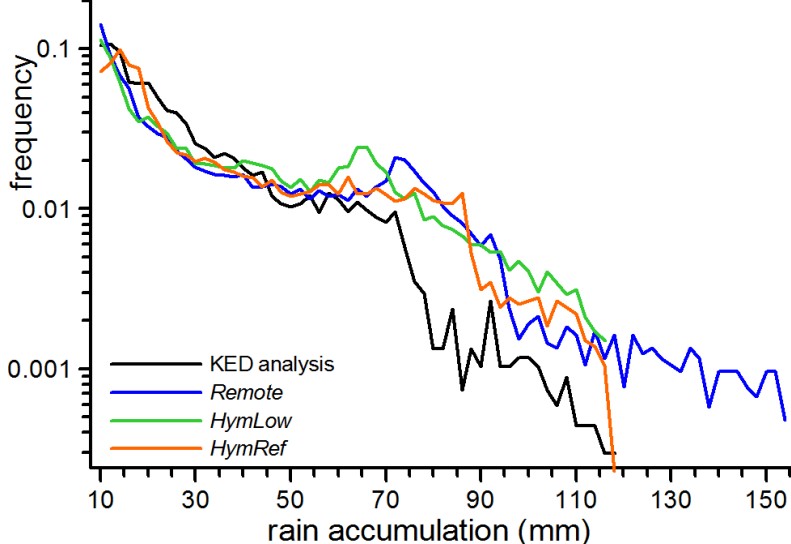

**Figure 7.** Normalized occurrence of observed and simulated rain accumulation from 6:00 ó 11:00 UTC. The bin size for the rain accumulation is 2 mm. Note the logarithmic frequency-axis.


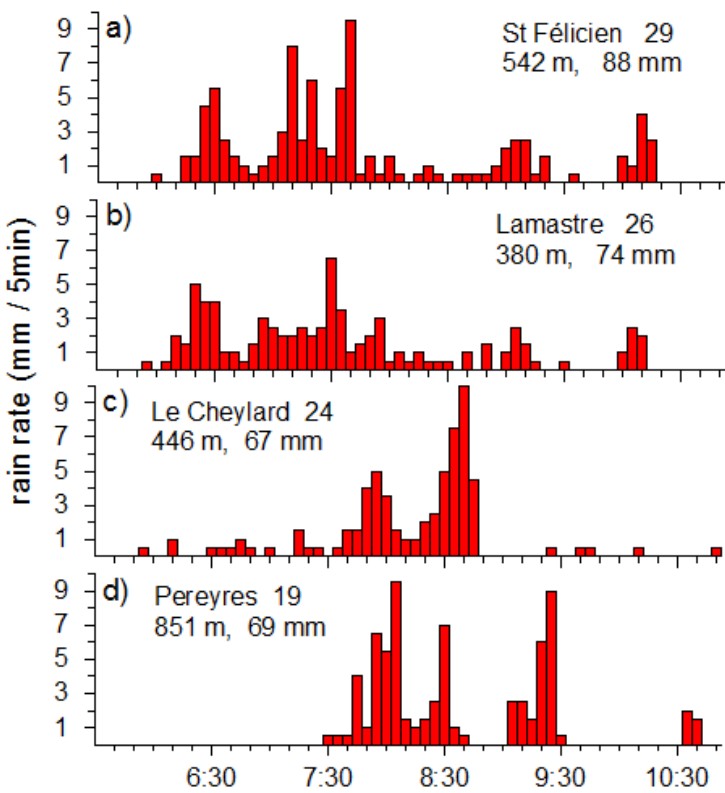

**Figure 8.** Temporal evolution of the 5 min rain rates of four observational points in the Vivarais for 26 September 2012.
Numbers behind the names give their location, as indicated in Fig. 2, as well as the elevation asl and the accumulated rain
amount.





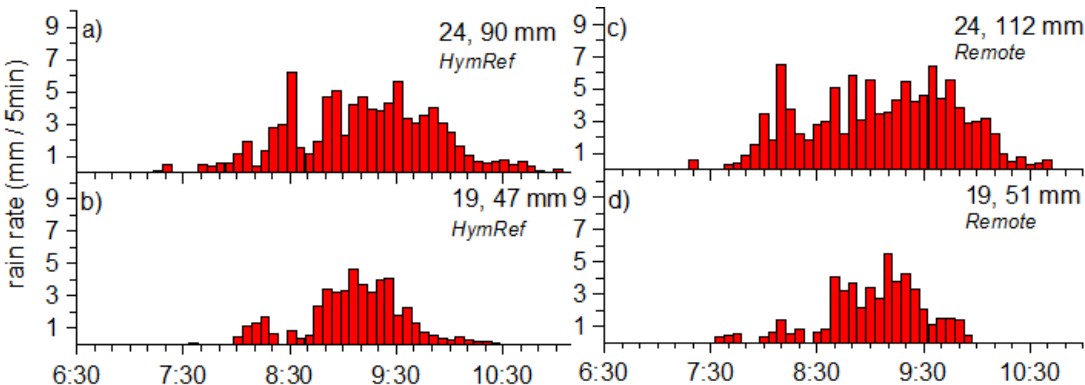

**Figure 9.** Time evolution of the simulated 5 min rain rates of IOP7a for scenarios *Remote* a) and b), and scenario *HymRef* c)

5    an d). Model results were taken for rain gauge locations 19 and 24 displayed in Fig. 2.




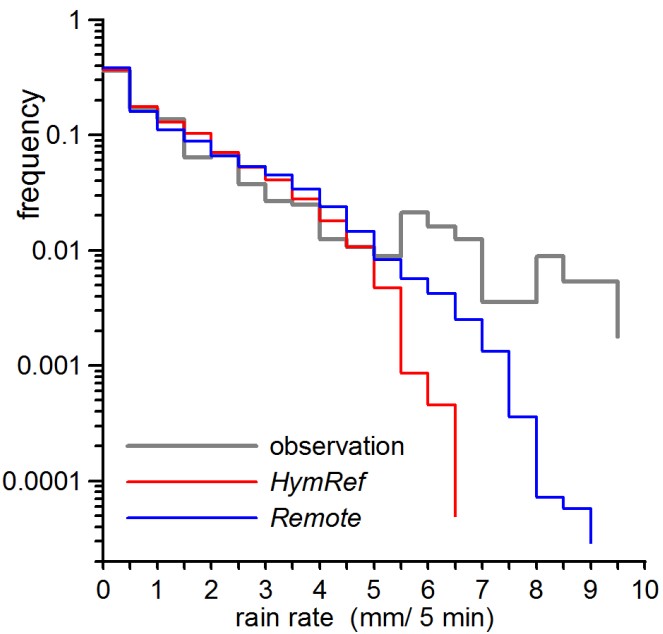

**Figure 10.** Normalized occurrence of the 5 minutes rain rate determined from 31 rain gauge observations (black line) and simulations in the third domain for *HymRef* (red line) and *Remote* (blue line).


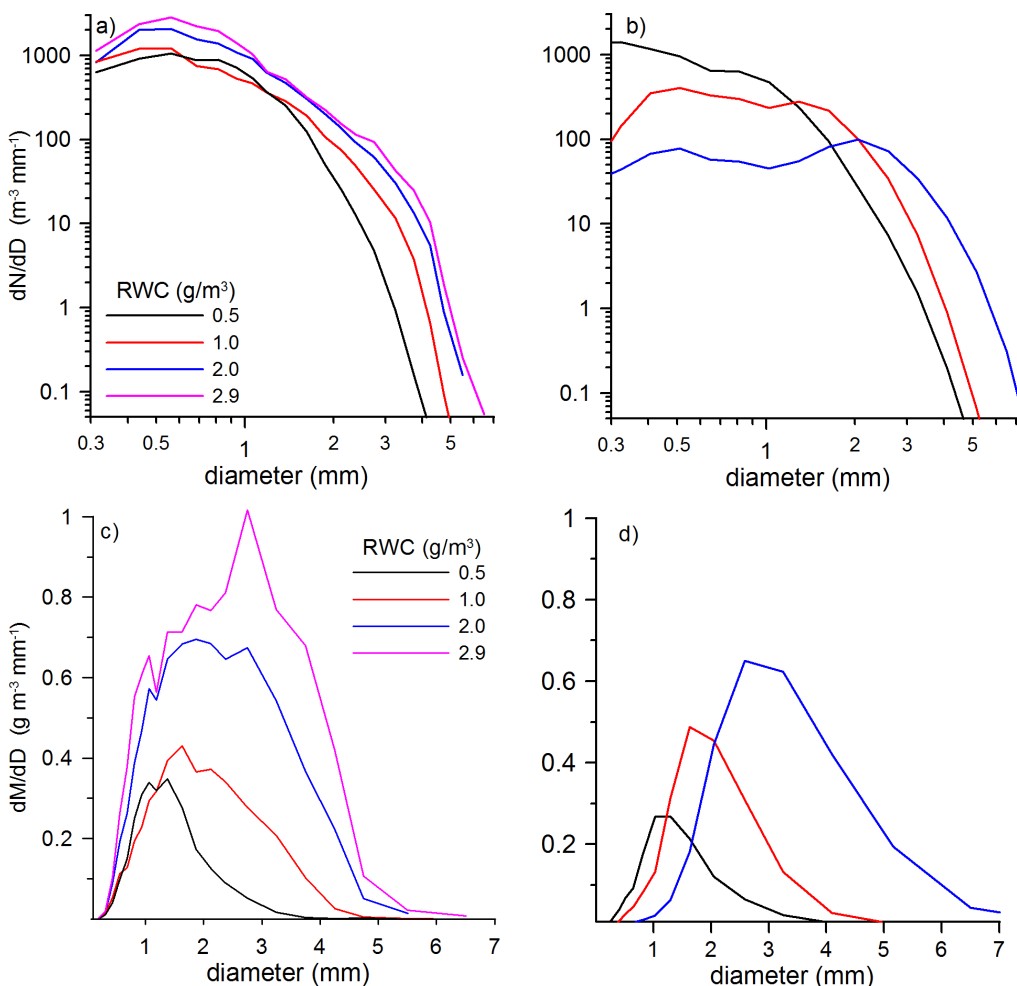

**Figure 11.** Number distribution functions of raindrops: (a) observed at La Souche in 950 m asl., (b) modelled for scenario *HymRef*. The equivalent mass distributions for the observed spectra are given in c), for the modelled spectra in d). The spectra were selected in four rainwater categories; each can deviate ± 20% from the given RWC value.

5|


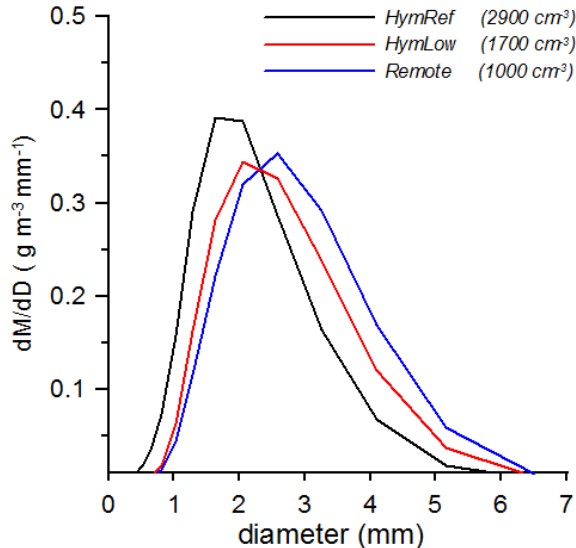

**Figure12.** Mean mass distributions of raindrops simulated for all surface elevations between 500 to 600 m asl in the third

5   model domain.