# Peer review of "The sensitivity of intense rainfall to aerosol particle loading - a comparison of bin-resolved microphysics modelling with observations of heavy precipitation from HyMeX IOP7a"

_Natural Hazards and Earth System Sciences, 2019_

## Referee Comment (RC1) · Anonymous Referee #1 · 15 Nov 2019

Review of nhess-2019-321:

*The sensitivity of intense rainfall to aerosol particle loading – a comparison of bin-resolved microphysics modeling with observations of heavy precipitation from HyMeX IOP7a*

**Overview**

The authors present simulations of a heavy precipitating event (IOP 7a) that occurred on Sept. 26, 2012, during the HyMeX observation campaign. Simulations are performed using the DESCAM bin microphysics scheme, which provides a detailed representation of hydrometeors, aerosols, and their interactions. The purpose of this paper is first to evaluate the rainfall prediction using this bin scheme, and then to assess the sensitivity of rainfall prediction to the initial aerosol loading taking advantage of the scheme's abilities.

Although we are far from being able to use such a detailed and costly microphysics scheme in operational NWP, this study is interesting both to advance our understanding of rain production processes, and to maybe serve as a reference for the development of simpler, less computationally expensive schemes.

The paper has a usual but efficient structure, and shows that the rainfall predicted using DESCAM agrees well with the observations overall, although some discrepancies remain both on the regional scale (location, intensity and timing of rain) and the raindrops scale (particle size distributions). Results also clearly show that changing the aerosol population has an impact on rainfall forecasts.

However, I think that the paper suffers from the issues raised below as "major comments" (minor comments are listed afterwards).

**Major comments**

- This study is conducted based on rainfall characteristics at the ground only. This is a deliberate choice from the authors, who mention twice that in-cloud features will be presented in a future work. Following this idea, the paper neither discusses the quality of the simulated convective system macroscopic and microscopic characteristics (cloud height, anvil extension, cloud composition) nor investigates the rain formation processes (eg. warm phase vs. mixed phase formation). This is especially lacking since the bin scheme is not expected to be used in operational NWP in the near future, but instead a very good tool for process studies and understanding. Thus, to me, the paper has more value as an introduction for a detailed cloud composition and processes study, despite being presented as a standalone paper. I can imagine a detailed study of the cloud composition and processes needs a paper by itself, so maybe the two parts could be made into a two-part paper (part 1 for model description and rainfall evaluation, part 2 for microphysics and processes) ?

- The paper sometimes stops short of providing or verifying an explanation for the presented results.
  - P6 l23-27: If the reach of the X-band radar is too short, why don't you use the french radar mosaic instead (especially since the radar is only used to check the large scale characteristics of the convective line) ?
  - P8 l6-10: The three simulations do not represent the precipitating system shift. Other studies of this case are mentioned in the paper (eg p3 l23, Hally et al 2014). How does that (or its consequence on the total rainfall amount slight mislocation) compare to others ? This could hint at the influence of large scale conditions used for coupling.

- P8 l25-30 & p9 l1-5: Various studies of the impact of aerosols on clouds and precipitation show different effects. Sometimes an increase in aerosol concentrations leads to reduced precipitations, sometimes to "convective invigoration" instead. What processes (is the impact on droplets or ice crystals concentrations more important ? More cloud droplets subjected to contact freezing with aerosols ?) are important in this specific case (organized, long-lasting convection, with an orographic forcing), and are they different from what was found for isolated convective cells ?
- P9 & fig 7 : The three simulations show very little differences below 70mm. Is there a physical explanation to the fact that aerosols only impact the occurrence of high precipitation ?
- P10 l11-18: The model resolution can explain some differences with the observations, eg. higher 5-min rainfall in the observations. However, there are differences between figs. 8 (observations) and 9 (models) that probably cannot be explained by the smoothing effect of the model resolution. The progressive increase and decrease in precipitation in the simulations, occurring over 20min to 1h, is more probably linked to differences in the convective system characteristics or dynamics. Is this linked to convective cells dissipating slower in the model ? To convective cores being surrounded by larger regions of moderate precipitation in the model than in the observations ?
- P11 l24+: To explain differences in rain size distributions between the model (at altitudes of 900-1000m) and observations (at 950m), the authors state that the cloud base may have been lower than simulated (1300 to 1400m). Are there no observations from the HyMeX campaign (Lidars ? Cloud radars ? Maybe MRRs ?) to support this, even if they were not located at the same place ?

- The three simulations use realistic aerosol loadings (Table 2) coming from observations for this specific case (*HymRef*), the cleanest HyMeX case (*HymLow*) which still has high aerosol concentrations, and another set of observations to represent cleaner conditions (*Remote*). The total number concentration for these simulations is, as stated by the authors, lowest in the *Remote* case and highest in the *HymRef* case, and therefore conclusions are drawn throughout the paper about the impact of an increase / decrease in aerosol number concentration. However, if we look only at aerosol modes 2 and 3 (because the smallest aerosols from mode 1 with a diameter around 0.05 microns are much harder to activate into cloud droplets or ice crystals), the number concentrations are highest in the *Remote* simulation (which also has the largest diameters for these two modes) and lowest in the *HymRef* simulation, so maybe the conclusions based on aerosol number concentrations could be reversed ? Simulations using the population from *HymRef* but modulated by the same factor for the 3 modes would make this conclusion easier. Maybe this can be clarified in the current simulations, through an analysis of the activation of smaller aerosols in the three simulations (total number of activated aerosols, activation height or temperature or timing for the different modes, …) ?

- Comparisons of raindrop size distributions show that the number of small rain drops at the ground is not very well represented by DESCAM. Although the distance from cloud base changes the shape of the rain PSD, authors state that the decrease in small drops numbers with an increase in rain water content was not observed by disdrometers at lower altitudes. This calls for some more detailed analysis: how is the drop PSD changing with height in the model vs. observations (Micro Rain Radars were deployed during HyMeX and provide the rain PSD at different heights, polarimetric radars can also help assess the rain characteristics ) ? Is this really possibly linked to the collisional break-up as suggested, or is this also possibly linked to overestimated collection rates, or errors in the sedimentation process ?

**Minor comments**

- p1,l19-21: add references for the climatology of extreme events in the region and modelling difficulties

- p1,l22-24: A lot more than just rain gauges and radars were available during the HyMeX campaign.
- P2,l24-27: The first stated objective is to show added value of a bin scheme vs. bulk schemes, however no result in the paper ever discuss the performance of bulk schemes. This should be moderated as there is no evidence of it in this paper → *the first objective of the paper is to evaluate the performance of a 3D mesoscale model including a bin microphysics scheme in predicting heavy rainfall.*
- P4,l30-34: Please precise the aerosol concentration decrease in the first 3km (how many aerosols remain at 3km and above?)
- p5,l1-2: Please precise if soluble aerosols act as CCN only, or can also act as INPs (eg. by immersion freezing)?
- P5,l7 & 13: some characters do not display correctly
- P6,l1-2: this sentence is not necessary as the flight date and location were already mentioned before.
- P6,l18: figure 4 is used in the text before figure 3?
- p7,l29: text mentions precipitation over 10mm, while fig.4a shows reflectivity in dBz.
- P8,l13-15: Is the Taylor diagram necessary for only 3 simulations?
- P9,l6-11: "considerable similarity" is exaggerated. For most precipitation accumulations, there is more difference between any simulation and observations, than between different simulations.
- P10,l19-26: say at the beginning of the paragraph that we are now looking at fig. 10 (I initially thought that the comment was not fitting the figure because I was still looking at fig. 9 that does not show 5min precipitation over 6 or 7mm for *Remote* simulation)
- p11,l6-8: What fraction of observed DSD spectra is ignored?
- P11,l23: between 900̶0̶ and 1000m
- p12,l3-11: mass distributions from fig. 12 are very similar. Can the small differences be explained by the differences on rain water content (especially for *Remote* which has a higher mean RWC, but also for *HymLow* which has the same mean RWC but maybe more extreme values), or is there also a difference in distributions at the same given RWC for the 3 simulations ?
- P14,l15: strange characters around "broken"
- Fig1: legend missing for the gray contours.

---

## Referee Comment (RC2) · Anonymous Referee #2 · 22 Dec 2019

**The sensitivity of intense rainfall to aerosol particle loading - a comparison of bin-resolved microphysics modelling with observations of heavy precipitation from HyMeX IOP7a**

Christina Kagkara, Wolfram Wobrock, Céline Planche and Andrea Flossmann

**General comments**

The study presents simulations of heavy precipitation event occurred on 26th Sept., 2012 during the HyMeX campaign. The simulations use the DESCAM bin-microphysical scheme which solves for changes in hydrometeors size distributions, including aerosols mas within drops and ice crystals.

One of the major objective of the study is using the detailed bin-resolved microphysical scheme to reproduce quantitatively precipitation characteristics such as: rain accumulation, rain size distribution, spatial and temporal variability - all compared to the observed rain gauges, distrometers and ground radars. Another objective is to check the sensitivity of surface precipitation and raindrop spectra to aerosol number concentration.

Overall the paper structured sufficiently well and present the potential of a bin-scheme in reproducing precipitation characteristics. However, the study suffers from some major deficiencies that force major revision.

**Specific comments**

The specific comments below refer to the following Major comments bullets:

- The study employed the DESCAM bin scheme but the manuscript is short in description how the bin scheme is doing better compared to previous studies that used 1M/2M/3M bulk microphysics (at least qualitatively). This should be (at least) included in the discussion section.

- There is a serious lack in physical argumentation for several deficiencies in the model results and lack of proper references for the cloud-precipitation-aerosol interaction work done by the community. There is also lack of vertical cloud structure (tendencies or budget analysis) that could links/leads to the surface rain characteristics that is the focus of the paper.

- There is a systematic problem in using the three test cases (HymRef, HymLow and Remote) to deduce the sensitivity to aerosols number concentration. In order to test the sensitivity to aerosol concentration, the systematic way would be to change the concentration of a certain mode of the same aerosols size distribution (ASD). The concentrations in the ASDs modes as shown in the paper differ substantially, which leads to different physical response of the clouds. Otherwise, if the authors stored the number of drops nucleated as function of supersaturation and/or as a function of the ASD dry size ranges -- this would be a preferable approach to isolate the

aerosols effect. In case this is not available, they would need to rephrase their conclusions as far as the aerosol sensitivity is concern. This is further required because they do not present the corresponding vertical cloud structure to help assessing certain deficiencies and/or aerosol sensitivity.

© Pg. 2 lines 2-3: There is no indication which microphysical schemes were used in the referenced papers. (minor)

© Pg. 2 lines 5-6: if previous studies ''succeed'', why were there significant differences in location, intensity and microphysical characteristics? They ''succeed'' according to what standard?

© Pg. 2 line 15: Again, agree in what sense? Did they use large temporal and spatial averaging technique? Is this sufficiently good? I would argue that any reasonable microphysical scheme can be compared to observations to some extent. In that case, why do we invest time to calculate spatial and temporal changes of hydrometeor spectra?

© Pg. 2 lines 20-24: In addition to mentioned above, you might want to stress that 1M/2M/3M bulk schemes have much more tuned microphysical processes / parameters, where bin microphysics have very few constraints apart from discrete grid for hydrometer mass/size into bins. See a comprehensive review in Khain et al. (2015).

© Pg. 2 line 32 – Pg. 3 line 2: Well, there is substantially larger amount of work being done in the cloud physics community than mentioned here. Please read (at least) the following references (and the references within) for a more complete 3D cloud-aerosol-precipitation interactions studies: Lynn et al. (2016), Marinescu et al. (2016), Fan et al. (2018), Marinescu et al. (2018), Shpund et al. (2019a, 2019b).

© Pg. 3 lines 29-30: This needs to be justified as the homogenous nucleation level and/or stratifom parts can easily get to 12-12.5 km easily. In addition, it is probably a way to reduce the computational loading, is this means the interaction between the outer-most and the inner domain are one-way? This should be clearly written.

© Pg. 4 lines 1-2: It looks the DESCAM scheme calculates the aerosol mass dissolves within drops and ice crystals. Within the cloud microphysics community, it is debatable if this worth the additional calculation loading. In part, this is why most of the modeling work uses this method only in warm clouds and/or idealize setup. Apart from the calculation loading, can you comment on how significant is it to your simulations, facing your goals to improve the precipitation characteristics?

© Pg. 4 line 5: Again, facing your goals the reader should understand how main features of the microphysical scheme works. Rain drops of 10mm are extremely rare (some thinks they just do not exist); as such it is important to understand how the scheme handles these potentially numerical artifacts of very large rain drops that aren't stable. This affects for sure your rain size distribution.

© Pg. 6 line 18: Why do you start with describing Figure 4? (minor)

© Pg. 6 line 20: should be sixth moment, not "sixth momentum"? what do you mean in "normalized" here? (minor)

© Pg. 7 line 3: can you please explain from physical perspective what prevent the model (dynamical core + microphysical scheme) from being able to reproduce the change in orientation?

© Pg. 7 line 16-19: can you please comment from the microphysical scheme perspective -- why the area of rain accumulation is different, especially the area of the accumulated rain of ~38 (mm) and below is significantly underestimated. It looks like the scheme (or the setup) has problems in simulating shallow convection and/or startiform clouds.

As a follow up query, how was the corresponding forecast of the 1M/2M/3M bulk microphysical schemes? Could you please comment on that.

© Pg. 7 line 30: Indeed, but the reader may ask himself what in the microphysical scheme lead to this changes? If the paper would have a more coherent microphysical analysis (vertical cloud structure) you would be able to explain that from physical point of view.

In addition, this is an example to the systematic problem in the ASD setup, where the Remote setup has 600 #/cm3 and 250 #/cm3 in the accumulation and the coarse modes, respectively. These modes are readily nucleate to droplets in any typical deep convection systems, and should lead to early rain formation (especially the coarse mode with 250 #/cm3). This is quite different aerosols regime.

© Pg. 8 lines 10-11: Have you checked your low-surface "cold pool"? The question is whether the limited spatial changes in rain results from a dynamical reason or underestimation in low-level rain amount and sizes which limits the evaporation and thus decrease the "cold pool". This is related to the large scale forcing vs. local convective instability.

© Pg. 9 lines 4-5: Regarding your conclusion that "more rain occurs when low particle number prevails" – this is likely to be true for 2 ASDs with different number concentration per modal size in a warm convective system. When you convolve number concentration between ASD modes, the rain can be initiated from different level in the cloud. As the convection becomes deep enough, lower CCN size penetrates to areas where high vertical velocity occurs and thus higher supersaturation above liquid/ice occurs (Sw, Si), and more smaller drops nucleates, which means more vapor is extracted from the atmospheric column (Si > Sw) compared to nucleation at lower levels where Sw is limited by warm rain; this serves as positive feedback that intensify the convection as more drop freezes at higher levels, as well as lead to increase in large/dense hydrometeor size which sediment and force downdraft and further positive feedback.

The above is called 'convective invigoration' that leads to more intense rain rate. In your Remote ASD setup you are not only reducing total aerosol number concentration, but you also "pushes" the clouds to rain-out (warm-rain) substantially earlier due to the increased number concentration in the accumulation and especially the coarse modes. Therefore, based on the limited analysis presented here, your conclusion needs to be rephrased to include the information about the differences in ASDs modal concentration

© Pg. 9 lines 12-13: again, you have forced at least 2 more degrees of freedom in that the Remote ASD has substantially different aerosol number concentration distributed between the modes. You need to address this by dedicated sensitivity test, or at least restrict your conclusions.

© Pg. 10 line 24: the value of 9mm / 5min in Figure 9 cannot be seen. Please comment or correct this value. (minor)

© Pg. 11 lines 25-30: There is no indication of the temperature near the surface and where is the freezing level placed. There is no clear indication how the averaging has been performed (space-wise). Since the model simulations clearly underestimates the area of ~25 mm (and below) and the averaging was made between 900-1000m, it is not clear to me how the RWC = 0.5 g/m3 rain size distribution in Figure 11b are reasonably compared to observations (as shown in Figure 11a). Such RWC are largely in the underestimated area and can be attributed to shallow convection or even to heavily stratiform precipitating clouds. Can you explain this apparent discrepancy?

Furthermore, in principle raindrops grows at the expense of small-medium size raindrops (0.3 – 1.5 mm) as these fall through the cloudy area, but this is quite a simplistic point of view as observations (Figure 11a) indicates that other processes are likely to be responsible for the ongoing supply of these small-medium raindrop near the surface and for vast range of RWC. These process are being determined well above the surface (for instance: melting process; breakup of large raindrop). Thus, based on this simplistic microphysical analysis made here, the conclusion drawn should be very careful as probably the model has some drawbacks in this aspect.

Pg. 12 line 22: what is the context for "superficial" here? (minor)

**Technical comments**

© Pg. 2 line 27: There is no meaning in "bin resolved"; you probably mean "size resolved".

© Pg. 3 line 26: Maybe "outermost model domain" is preferable. Also, the resolution increases, where the grid spacing decreases.

© Pg. 4 line 1: A microphysical scheme (like the DESCAM) calculates (or prognoses) the temporal and spatial changes in the distribution functions. The overall set up of the dynamical core coupled to the microphysical scheme with the BC/IC "*simulates*" a particular test case and the corresponding fields (rain, CWC, RWC, etc.).

© Pg. 4 line 26: … a third *aerosol distribution* with lower *number concentration* is used.

© Pg. 4 line 33 (and throughout the text): number distribution is confusing. Use number concentration or/and aerosol/droplet/rain size distribution.

© Pg. 11 lines 3: rain size distribution should be noted as RSD and not DSD. DSD is droplet/drops size distribution.

**References:**

Khain, A.P., Beheng, K.D., Heymsfield, A., Korolev, A., Krichak, S.O., Levin, Z., Pinsky, M., Phillips, V., Prabhakaran, T., Teller, A. and van den Heever, S.C., 2015. Representation of microphysical processes in cloud-resolving models: Spectral (bin) microphysics versus bulk parameterization. *Reviews of Geophysics*, *53*(2), pp.247-322.

Lynn, B. H., Khain, A. P., Bao, J. W., Michelson, S. A., Yuan, T., Kelman, G., … Benmoshe, N. (2016). The sensitivity of Hurricane Irene to aerosols and ocean coupling: simulations with WRF spectral bin microphysics. *J. Atmos. Sci.*, *73*(2), 467–486.

Marinescu, P. J., Heever, S. C., Saleeby, S. M., & Kreidenweis, S. M. (2016). The microphysical contributions to and evolution of latent heating profiles in two MC3E MCSs. *J. Geophys. Res., Atmospheres*, *121*(13), 7913–7935. doi.org/10.1002/2016JD024762

Marinescu, P.J., van den Heever, S.C., Saleeby, S.M., Kreidenweis, S.M. and DeMott, P.J., 2017. The microphysical roles of lower-tropospheric versus midtropospheric aerosol particles in mature-stage MCS precipitation. *Journal of the Atmospheric Sciences*, *74*(11), pp.3657-3678

Fan J., D. Rosenfeld, Y. Zhang, S. E Giangrande, Z. Li, Luiz AT Machado, S. T Martin, Y. Yang, J. Wang, P. Artaxo, H. MJ Barbosa, R. C Braga, J. M Comstock, Z. Feng, W. Gao, H. B Gomes, F. Mei, C. Pöhlker, M. L Pöhlker, U. Pöschl, R. AF de Souza (2018): Substantial convection and precipitation enhancements by ultrafine aerosol particles. *Science* 359 (6374), 411-418

Shpund, J., Khain, A. and Rosenfeld, D., 2019. Effects of Sea Spray on the Dynamics and Microphysics of an Idealized Tropical Cyclone. *Journal of the Atmospheric Sciences*, *76*(8), pp.2213-2234.

Shpund, J., Khain, A., Lynn, B., Fan, J., Han, B., Ryzhkov, A., Snyder, J., Dudhia, J. and Gill, D., 2019. Simulating a Mesoscale Convective System Using WRF with a New Spectral Bin Microphysics-Part 1: Hail vs Graupel. *Journal of Geophysical Research: Atmospheres*.

A. P. Khain, V. Phillips, N. Benmoshe, A. Pokrovsky, J. Atmos. Sci. 69, 2787–2807 (2012)

---

## Author Comment (AC1) · 10 Feb 2020

**NHESS-2019-321**
**Responses to reviews**

The sensitivity of intense rainfall to aerosol particle loading - a comparison of bin-resolved microphysics modelling with observations of heavy precipitation from HyMeX IOP7a

C. Kagkara, W. Wobrock, C. Planche and A. I. Flossmann

7 February, 2020

**Note to the editor**

We thank all of the reviewers, whose comments have led to significant improvements in the analysis and our manuscript. Each question and remark of the reviewer is answered below point by point.

Changes in the manuscript and the reply to the individual remarks of the reviewers are marked in red for easier notice.

We would like to point out, however, that our choice of NHESS as publication journal has motivated our focus on the study of the surface precipitation over a region, which is often affected by flash floods. Following the request of the reviewer we have added some more discussion on in-cloud processes, however the in-depth analysis of the cloud microphysics and their comparison with the available airborne probes will be published in another more appropriate journal.

**Responses to reviewer #1's comments**

Answers to **Major comments**

- This study is conducted based on rainfall characteristics at the ground only. This is a deliberate choice from the authors, who mention twice that in-cloud features will be presented in a future work. Following this idea, the paper neither discusses the quality of the simulated convective system macroscopic and microscopic characteristics (cloud height, anvil extension, cloud composition) nor investigates the rain formation processes (eg. Warm phase vs. mixed phase formation). This is especially lacking since the bin scheme is not expected to be used in operational NWP in the near future, but instead a very good tool for process studies and understanding. Thus, to me, the paper has more value as an introduction for a detailed cloud composition and processes study, despite being presented as a standalone paper. I can imagine a detailed study of the cloud composition and processes needs a paper by itself, so maybe the two parts could be made into a two-part paper (part 1 for model description and rainfall evaluation, part 2 for microphysics and processes)?

In order to provide the reader with a better description of the characteristics of the macroscopic cloud system (cloud height, vertical cloud composition) we added in chapter 4 two vertical cross sections indicating IWC and RWC as well as temperature and humidity conditions for the cloud system. This also clarifies several individual questions of both reviewers.
The inserted text:

The vertical structure of the simulated cloud and rain field is illustrated in Figs. 6a and b. Both figures show the same vertical cross section for the innermost domain reaching from the southern border (at x=529, y=560 km) to the northern limit (at x=579, y=688 km). Fig. 6a gives the ice water content (IWC), Fig. 6b the rainwater content RWC for values larger 0.1 g/m$^3$. For the calculation of the RWC from the modelled drop size distribution only drop sizes larger 100 µm were considered. The illustration Fig.6b shows a quite continuous rain field during the intense rain episode at 8:20 h. Important RWC of 2-2.5 g/m$^3$ mainly forms close to the melting level. The 0°C levels varied due to the strong vertical motion over the complex terrain between altitudes from 3.3 and 3.7 km. We can also detect in Fig. 6b that raindrops appear in elevated layers up to -20°C. The IWC, however, reached much higher altitudes but the presences of ice values larger than 1 g/m$^3$ rarely exceeded a height of 8 km, which is in agreement with aircraft in-situ and cloud radar observations performed during the same time period. The illustration of the field of IWC indicates that the cloud system mainly developed to mid-tropospheric layers and convection did not exceed 7-8 km. Thus, the tropopause level could not be attained and consequently no anvil formation took place. Fig. 6a also includes two contour lines for relative humidity of 90% and 98%. The high humidity in the lower layers is caused by the southern flow from the nearby Mediterranean Sea. Relative humidity of 90% appears around 1000 m asl, 98% 200 to 300 m above. Cloud base height, i.e. the formation of cloud droplets is located at altitudes around 1200-1300 m.

The formation of the convective system was triggered by orographic lifting over the Cevennes Vivarais Mountains. The rapid cloud formation and intensification was in addition favoured by the high vapor loading in the lower atmospheric layers, arriving from the warm Mediterranean Sea and persisting for several hours.

- The paper sometimes stops short of providing or verifying an explanation for the presented results. We tried to justify all our conclusions.

o  P6 l23-27: If the reach of the X-band radar is too short, why don't you use the French radar mosaic instead (especially since the radar is only used to check the large scale characteristics of the convective line)?
The data from the nearby weather radars Bollène and Nîmes, available from the HyMEx data base, are not corrected for ground clutter and attenuation. As we don't have the competence to do these corrections, we excluded a further comparison with the model results. Corrected ARAMIS radar data (as a composite) only were available as surface rain. These data were used by the KED technique to determine (combined with rain gauges) the hourly rainfall, which we finally used for our comparison with modeled rain parameters. Thus, the radar based large scale characteristics are included and discussed for the precipitation field but not for radar reflectivity in the atmospheric levels above.

o  P8 l6-10: The three simulations do not represent the precipitating system shift. Other studies of this case are mentioned in the paper (eg p3 l23, Hally et al 2014). How does that (or its consequence on the total rainfall amount slight mislocation) compare to others? This could hint at the influence of large scale conditions used for coupling.

Hally et al (2014) investigated the precipitation event on larger scales. Hourly rainfall, averaged over an area of 400x 400 km$^2$, was compared to observation. Individual changes in the evolution of the cloud system were not considered in their study.

- o  P8 l25-30 & p9 l1-5: Various studies of the impact of aerosols on clouds and precipitation show different effects. Sometimes an increase in aerosol concentrations leads to reduced precipitations, sometimes to "convective invigoration" instead. What processes (is the impact on droplets or ice crystals concentrations more important? More cloud droplets subjected to contact freezing with aerosols?) are important in this specific case (organized, long-lasting convection, with an orographic forcing), and are they different from what was found for isolated convective cells?

As both reviewers pointed out that hydrometeor formation by aerosol particles may be also important in elevated cloud layers due to "convective invigoration", we will give a short explanation from our point of view (These considerations will not be part of the results presented in the paper, as considered out of scope).

Aerosol particles and especially water vapor are abundant in the lower atmospheric layers were the cloud forms. In our case of IOP7a the water vapor mixing ratio next to cloud base is about 10 g/kg. It is this water vapor in the lower 1000 m which is responsible for cloud formation and the subsequent cloud evolution over several kilometers in altitude. The convection (vertical motion) which is triggered by the strong heat release above cloud base, transports vapor, drops and aerosol particles to higher levels. Supersaturation gets strong in the first 3-4 km above cloud base and thus most nucleation of aerosol particles to drops takes places in this stage. Ice crystal formation may occur in our case from 4 km upward when temperature decreases below -3 to -5°C. The heterogeneous nucleation rates, however, are quite weak and ice particle formation by nucleation of non-activated particles remains low, even up to 6-7 km, when temperatures are higher than -15 to -20 °C. Ice formation occurs in this temperature range, but crystals form to a large extent by condensational freezing and Bigg freezing of already existing drops.

When rising to higher levels until -28°C (homogeneous nucleation will start for T< -28°C) ice supersaturation can become more than 120 % and ice nucleation rate strongly increases. Invigoration of convection can arise at these altitudes (and also in higher levels for deeper cloud systems), when vertical momentum and water vapor are advected.  But aerosol – hydrometeor interactions are only insignificantly affected for several reasons:

(1) the number concentration of the aerosol particles in the elevated levels is low compared to the cloud base environment

(2) the remaining interstitial concentration of particles (which can serve as CCN and INP) present in the raising updraft is also low as most of them were already consumed for drop formation in lower altitudes

(3) the pathway to form new droplets from non-activated aerosol is negligible due to the reduced kinetics of the water vapor diffusion for temperature < -15°C. (i.e. the Köhler equilibrium fails)

(4) even if ice supersaturation is really high, we have to be aware that the supply with water vapor is quite low in these altitudes. In our case study of IOP7a water vapor mixing ratio is 1-2 g/kg at -20°C. From the new Fig. 6a we can see that maximum simulated (and also observed) IWC in this level can be well above 2 g/m$^3$ which correspond to an ice mixing ratio of 3.6 g/kg. Thus, the environmental condition cannot be responsible for the high ice mass and also crystal concentration encountered in these elevated levels. Their presence is not a consequence of new ice nucleation and ice growth but due to advection

from the lower altitudes where droplets form and grow in a significant way and freeze in the elevated levels.

In summary, based on our actual knowledge on droplet activation and ice nucleation we cannot see a specific effect during convective invigoration on these processes.

o  P9 & fig 7: The three simulations show very little differences below 70mm. Is there a physical explanation to the fact that aerosols only impact the occurrence of high precipitation?

We agree, there are very little differences in the relative occurrence of surface rain accumulation between 10 to 70 mm for all three scenarios. This may result from the overall similarity in the thermodynamic / dynamic evolution for the three cases. But we have to keep in mind the "relative" character of the frequency distribution of Fig.7 (now Fig.9), which probably does not allow concluding unambiguously that *aerosols only impact high rain accumulations*. All scenarios have spatial differences in surface rain coverage and intensity: the number of surface grid points with rain accumulation from 10 to 70 mm are in the *Remote* case 39210, but only 1720 in the *HymRef*. Consequently also total accumulated rain in the *Remote* case is significantly higher in the range from 10-70 mm. This becomes also clear when comparing the total mass of rainwater in Table 2. Subtracting results for Rain amount (>10mm) from Rain amount (>50 mm) gives 31.5 mm for *HymRef* but 38.2 mm for *Remote*.

Thus, Fig.7 (now Fig.9) only allows unambiguously the conclusion, that low aerosol concentration favors the formation of strong to very strong rain accumulation. The physical reason for this is the higher supersaturation that develops in a cleaner atmosphere. The higher the supersaturation, the more water and ice can form.

▪  P10 l11-18: The model resolution can explain some differences with the observations, eg. higher 5-min rainfall in the observations. However, there are differences between figs. 8 (observations) and 9 (models) that probably cannot be explained by the smoothing effect of the model resolution. The progressive increase and decrease in precipitation in the simulations, occurring over 20min to 1h, is more probably linked to differences in the convective system characteristics or dynamics **(1)**. Is this linked to convective cells dissipating slower in the model **(2)**? To convective cores being surrounded by larger regions of moderate precipitation in the model than in the observations **(3).**

**(1):** Yes, it is to a certain extend right that the differences in local rain rate are due to the "differences in the convective system characteristics or dynamics". From the X-band radar comparison we could detect that the observed convective cells are patchier and their orientation more often deviate from the main horizontal flow. The simulated convective cells are mores steady state, especially over the ridge of Vivarais (between gauge 19 and 24 in Fig.3) where precipitation was strongest. But in addition this difference in dynamics is a consequence of non-resolved sub-grid effects given by our grid resolution of dx=dy=500 m.
**(2)**: We compared in the region covered by the X-band radar (thus over the Cevennes mountains) the formation and dissipation of individual cells with the modeled ones and found that convective cells formed and dissipated on the same time scales.
**(3)**: We also investigated the question if "this is linked to convective cores being surrounded by larger regions of moderate precipitation in the model than in the observations" by comparing modeled and observed X-band reflectivity. This comparison is illustrated by the frequency distribution in Fig. R1. We

can detect from this illustration, the model overestimates slightly the frequency of observed Z in the range from 27 to 37 dBZ, where moderate rain could be suspected. In the range from 19 to 27 dBZ, however, the model is strongly underestimating the weaker precipitation zones. From this comparison a clear statement concerning the reviewer's question is difficult, but we think this result indicates a reasonable agreement with the moderate precipitation zones (Z > 27 dBZ) and the overestimation of moderate precipitation by the model is quite weak and not the reason for the extended duration of rainfall after the transit of the main core.

[Figure]

**Figure R1.** Probability density function of the X-band radar observations from 06:40 to 10:40 UTC and modeled with DESCAM over the same period every 20 min

- P11 l24+: To explain differences in rain size distributions between the model (at altitudes of 900-1000m) and observations (at 950m), the authors state that the cloud base may have been lower than simulated (1300 to 1400m). Are there no observations from the HyMeX campaign (Lidars? Cloud radars? Maybe MRRs?) to support this, even if they were not located at the same place?

Measurements coming from MRRs and from the airborne cloud radar confirm that precipitation was reaching until the surface, but don't give a hint on cloud base locations. In addition the low quality of the MRR data did unfortunately not allow a reliable retrieval of the RSD.
Lidars were not running during this IOP in the region of the Cevennes-Vivarais.

- The three simulations use realistic aerosol loadings (Table 2) coming from observations for this specific case (HymRef), the cleanest HyMeX case (HymLow) which still has high aerosol concentrations, and another set of observations to represent cleaner conditions (Remote). The total number concentration for these simulations is, as stated by the authors, lowest in the Remote case and highest in the HymRef case, and therefore conclusions are drawn throughout the paper about the impact of an increase / decrease in aerosol number concentration. However, if we look only at aerosol modes 2 and 3 (because the smallest aerosols from mode 1 with a diameter around 0.05 microns are much harder to activate into cloud droplets or ice crystals), the number concentrations are highest in the Remote simulation (which also has the largest diameters for these two modes) and lowest in the HymRef simulation, so maybe the conclusions based on aerosol number concentrations could be reversed? Simulations using the population from HymRef but modulated by the same factor

for the 3 modes would make this conclusion easier. Maybe this can be clarified in the current simulations, through an analysis of the activation of smaller aerosols in the three simulations (total number of activated aerosols, activation height or temperature or timing for the different modes…)?

Yes, there is a typo in our Table 2. The diameters of mode 2 and mode 3 for the *Remote case* are a factor 10 smaller than indicated, and used in the calculations. We apologize for this inaccuracy.
We added in the article Fig.2, which displays the size distribution with a linear ordinate for the number concentration to better illustrate the differences between the 3 scenarios.

- Comparisons of raindrop size distributions show that the number of small rain drops at the ground is not very well represented by DESCAM. Although the distance from cloud base changes the shape of the rain PSD, authors state that the decrease in small drops numbers with an increase in rain water content was not observed by disdrometers at lower altitudes. This calls for some more detailed analysis: how is the drop PSD changing with height in the model vs. observations (Micro Rain Radars were deployed during HyMeX and provide the rain PSD at different heights, polarimetric radars can also help assess the rain characteristics)?

Indeed, this calls for more detailed analysis, which will be subject of another article focusing on the microphysical structure of the cloud field. Starting here with this subject would automatically demand for more explanation on the processes in the layers above and so on … and finally oversizing the paper.
As indicated before, rain PSD of Micro Rain Radars available during this event were not reliable or simply wrong (i.e. completely in contradiction to the disdrometer observations).
How can the use of polarimetric radars better assess rain characteristics? The PPI scans of the nearby weather radar all remained well below the melting level.

- Is this really possibly linked to the collisional break-up as suggested, or is this also possibly linked to overestimated collection rates, or errors in the sedimentation process?

Yes, we think that the treatment of breakup in the microphysics schemes starts to late, i.e. the large drops > 4mm can form, which leads, as you suggested to an overestimation of the collection rates with the small precipitation sizes.

**Minor Comments**
- p1,l19-21: add references for the climatology of extreme events in the region and modelling difficulties.
  We added reference to Sénési et al., 1996; Romero et al., 2000; Delrieu et al., 2005; Silvestro et al., 2012; Rebora et al., 2013

- p1,l22-24: A lot more than just rain gauges and radars were available during the HyMeX campaign.
  We wanted to introduce the main instruments exploited in this study. We modified the text in order to clarify that these types of instruments were not the only ones available during HyMeX.

- P2,l24-27: The first stated objective is to show added value of a bin scheme vs. bulk schemes, however no result in the paper ever discuss the performance of bulk schemes. This should be moderated as there is no evidence of it in this paper → the first objective of the paper is to evaluate the performance of a 3D mesoscale model including a bin microphysics scheme in predicting heavy rainfall.

We agree. The objective of this paper is not to compare the bin vs. the bulk representation of the microphysics but to evaluate the performances of a bin model. We clarified this in the text.

- P4,l30-34: Please precise the aerosol concentration decrease in the first 3km (how many aerosols remain at 3km and above?)

  The aerosol concentration decreases exponentially up to 3km height in a same way for all the different simulations. In HymRef, the concentration at 3 km is approx. 900 $cm^{-3}$. We clarified this in the manuscript.

- p5,l1-2: Please precise if soluble aerosols act as CCN only, or can also act as INPs (e.g. By immersion freezing)?

  The aerosol particles are assumed to be ammonium sulfate that is 40 % soluble with molecular weight of 132 g/mole and 60 % of insoluble silicates. They can act as INPs too. We clarified the manuscript.

- P5,l7 & 13: some characters do not display correctly

  We corrected that.

- P6,l1-2: this sentence is not necessary as the flight date and location were already mentioned before.

  We deleted the sentence.

- P6,l18: figure 4 is used in the text before figure 3?

  Figure 3a (now 4a) is used first - in the paragraph of the section 3. So we didn't change the order of the figures.

- p7,l29: text mentions precipitation over 10mm, while fig.4a shows reflectivity in dBZ.

  We corrected the manuscript where the referencing where wrong. It was Fig. 3a (now Fig. 4a).

- P8,l13-15: Is the Taylor diagram necessary for only 3 simulations?

  Even if only 3 simulations are performed, we preferred the Taylor diagram instead of a table because we thought it helps to compare the simulations with observations, especially to evaluate their performance.

- P9,l6-11: "considerable similarity" is exaggerated. For most precipitation accumulations, there is more difference between any simulation and observations, than between different simulations.
  We modified the text.

- P10,l19-26: say at the beginning of the paragraph that we are now looking at fig. 10 (I initially thought that the comment was not fitting the figure because I was still looking at fig. 9 that does not show 5min precipitation over 6 or 7mm for Remote simulation)

  We clarified the manuscript because in this paragraph, we compared the Figs 8 and 9 (now Figs 10 and 11).

- p11,l6-8: What fraction of observed DSD spectra is ignored?

  The disdrometer of La Souche provided 70 spectra each with a 1 min sampling time. We excluded 9 spectra leading to RWC from 4.9 to 7.6 $g/m^3$ (and rain rates from 120 to 200 mm/h). This was included in the text

- P11,l23: between 900 and 1000m

  Corrected

- p12,l3-11: mass distributions from fig. 12 are very similar. Can the small differences be explained by the differences on rain water content (especially for Remote which has a higher mean RWC, but also for HymLow which has the same mean RWC but maybe more extreme values), or is there also a difference in distributions at the same given RWC for the 3 simulations ?

  We know from this analysis and from previous studies that a cleaner atmosphere is able to develop larger cloud drop sizes and more liquid water. The objective of this illustration is to demonstrate that this size increase is also conserved in the mm size range.

  Fig. 12 (now 14) responds to a large extent to the second part of the question (is there a difference in distributions at the same given RWC). HymRef and HymLow have the same mean RWC, but drops smaller than 2 mm are more frequent in HymRef than in HymLow and for drops larger than 2.5 mm vice versa. And these differences will certainly be conserved when we restrict the Remote RSD to lower values of RWC.

- P14,l15: strange characters around "broken"

  We corrected the text.

- Fig1: legend missing for the gray contours.

  We clarified the manuscript.

---

## Author Comment (AC2) · 10 Feb 2020

**NHESS-2019-321**

**Responses to reviews**

The sensitivity of intense rainfall to aerosol particle loading - a comparison of bin-resolved microphysics modelling with observations of heavy precipitation from HyMeX IOP7a

C. Kagkara, W. Wobrock, C. Planche and A. I. Flossmann

7 February, 2020

**Note to the editor**

We thank all of the reviewers, whose comments have led to significant improvements in the analysis and our manuscript. Each question and remark of the reviewer is answered below point by point.

Changes in the manuscript and the reply to the individual remarks of the reviewers are marked in red for easier notice.

We would like to point out, however, that our choice of NHESS as publication journal has motivated our focus on the study of the surface precipitation. Following the request of the reviewer we have added some more discussion on in-cloud processes, however the in-depth analysis of the cloud microphysics and their comparison with the available airborne probes will be published in another more appropriate journal.

**Responses to reviewer #2's comments**

The specific comments below refer to the following Major comments bullets:

- The study employed the DESCAM bin scheme but the manuscript is short in description how the bin scheme is doing better compared to previous studies that used 1M/2M/3M bulk microphysics (at least qualitatively). This should be (at least) included in the discussion section.

    The objective of this paper is not to compare the bin vs. the bulk representation of the microphysics but to evaluate the performances of a bin model. We clarified this in the text.

- There is a serious lack in physical argumentation for several deficiencies in the model results and (1) lack of proper references for the cloud-precipitation-aerosol interaction work done by the community. There is also lack of vertical cloud structure (2) (tendencies or budget analysis) that could links/leads to the surface rain characteristics that is the focus of the paper.

Concerning point (1), we added more references on recent publications in the field of aerosol – cloud interaction. Point (2): In order to provide for the reader a better description of the characteristics of the macroscopic cloud system (cloud height, vertical cloud composition) we included in chapter 4 two

vertical cross sections indicating IWC and RWC as well as temperature and humidity conditions for the cloud system. This also clarifies several individual questions of both reviewers.

The inserted text:
The vertical structure of the simulated cloud and rain field is illustrated in Figs. 6a and b. Both figures show the same vertical cross section for the innermost domain reaching from the southern border (at x=529, y=560 km) to the northern limit (at x=579, y=688 km). Fig. 6a gives the ice water content (IWC), Fig. 6b the rainwater content RWC for values larger 0.1 g/m$^3$. For the calculation of the RWC from the modelled drop size distribution only drop sizes larger 100 µm were considered. The illustration Fig.6b shows a quite continuous rain field during the intense rain episode at 8:20 h. Important RWC of 2-2.5 g/m$^3$ mainly forms close to the melting level. The 0°C levels varied due to the strong vertical motion over the complex terrain between altitudes from 3.3 and 3.7 km. We can also detect in Fig. 6b that raindrops appear in elevated layers up to -20°C. The IWC, however, reached much higher altitudes but the presences of ice values larger than 1 g/m$^3$ rarely exceeded a height of 8 km, which is in agreement with aircraft in-situ and cloud radar observations performed during the same time period. The illustration of the field of IWC indicates that the cloud system mainly developed to mid-tropospheric layers and convection did not exceed 7-8 km. Thus, the tropopause level could not be attained and consequently no anvil formation took place. Fig. 6a also includes two contour lines for relative humidity of 90% and 98%. The high humidity in the lower layers is caused by the southern flow from the nearby Mediterranean Sea. Relative humidity of 90% appears around 1000 m asl, 98% 200 to 300 m above. Cloud base height, i.e. the formation of cloud droplets is located at altitudes around 1200-1300 m.

The formation of the convective system was triggered by orographic lifting over the Cevennes Vivarais Mountains. The rapid cloud formation and intensification was in addition favoured by the high vapor loading in the lower atmospheric layers, arriving from the warm Mediterranean Sea and persisting for several hours.

- There is a systematic problem in using the three test cases (HymRef, HymLow and Remote) to deduce the sensitivity to aerosols number concentration. In order to test the sensitivity to aerosol concentration, the systematic way would be to change the concentration of a certain mode of the same aerosols size distribution (ASD). The concentrations in the ASDs modes as shown in the paper differ substantially, which leads to different physical response of the clouds. Otherwise, if the authors stored the number of drops nucleated as function of supersaturation and/or as a function of the ASD dry size ranges -- this would be a preferable approach to isolate the aerosols effect. In case this is not available, they would need to rephrase their conclusions as far as the aerosol sensitivity is concern. This is further required because they do not present the corresponding vertical cloud structure to help assessing certain deficiencies and/or aerosol sensitivity.

In fact, there was a typo in our Table 1. The diameters of mode 2 and mode 3 for the Remote case are a factor 10 smaller than given. We apologize for this inaccuracy.
The new figure 2 gives the size distribution with a linear ordinate for the number concentration to better illustrate the differences between the 3 scenarios.

Regarding the sensitivity tests: in previous papers, we have done extensive sensitivity studies varying aerosol modes. The objective here was to study the potential response in precipitation to different actually observed pollution scenarios in the region.

© Pg. 2 lines 2-3: There is no indication which microphysical schemes were used in the referenced papers. (minor)

The following sentence was added:

In particular, the model studies done in the HyMex context applied the one moment ICE3 scheme (Pinty and Jabouille, 1998), the work of Tauffour et al. (2018) compares in addition with a two-moment scheme LIMA (Vié et al, 2016).

© Pg. 2 lines 5-6: if previous studies ''succeed'', why were there significant differences in location, intensity and microphysical characteristics? They ''succeed'' according to what standard?

The wording was changed.

© Pg. 2 line 15: Again, agree in what sense? Did they use large temporal and spatial averaging technique? Is this sufficiently good? I would argue that any reasonable microphysical scheme can be compared to observations to some extent. In that case, why do we invest time to calculate spatial and temporal changes of hydrometeor spectra?

The wording was changed.

© Pg. 2 lines 20-24: In addition to mentioned above, you might want to stress that 1M/2M/3M bulk schemes have much more tuned microphysical processes / parameters, where bin microphysics have very few constraints apart from discrete grid for hydrometer mass/size into bins. See a comprehensive review in Khain et al. (2015).

The text was changed and now reads : "One major objective of this study is, thus, to test if a bin resolved microphysics module in a 3D mesoscale model is able to reproduce a real case of intense precipitation using the dataset obtained during IOP7a of HyMeX which then can help in the future to improve the bulk models that often have difficulties to simulate intense precipitation, in particular regarding the rain maxima requiring alerts for the population, due to the constraints of the prescribed spectra (Flossmann and Wobrock, 2019)."

© Pg. 2 line 32 – Pg. 3 line 2: Well, there is substantially larger amount of work being done in the cloud physics community than mentioned here. Please read (at least) the following references (and the references within) for a more complete 3D cloud-aerosol-precipitation interactions studies: Lynn et al. (2016), Marinescu et al. (2016), Fan et al. (2018), Marinescu et al. (2018), Shpund et al. (2019a, 2019b).

We included references of recent publication from Marinescu et al. (2018) et Shpund et al. (2019a) dealing with similar scientific objectives.

© Pg. 3 lines 29-30: This needs to be justified as the homogenous nucleation level and/or stratiform parts can easily get to 12-12.5 km easily. In addition, it is probably a way to reduce the computational loading, is this means the interaction between the outer-most and the inner domains are one-way? This should be clearly written.

Information on the vertical extension of the cloud field was available from cloud radar observations above before the model was set up. This is now illustrated by the new Fig.6 displaying the field of IWC and rain as a vertical cross-section. All nesting interactions were treated in the "two way" mode. We added this information at the end of line 29.

© Pg. 4 lines 1-2: It looks the DESCAM scheme calculates the aerosol mass dissolves within drops and ice crystals. Within the cloud microphysics community, it is debatable if this worth the additional calculation loading. In part, this is why most of the modeling work uses this method only in warm clouds and/or idealize setup. Apart from the calculation loading, can you comment on how significant is it to your simulations, facing your goals to improve the precipitation characteristics?

A fundament of the microphysical modeling in DESCAM is to respect the presence of aerosol and cloud particles at the same time, since without aerosols, clouds would not form in most parts of the atmosphere. This also holds for precipitation formation. The condensed water mass (liquid or solid) present in a cloud is determined by the strength of the phase transition of water vapor. This process is measured by supersaturation, which depends not only from atmospheric dynamics but also from the presence of aerosol particles. Once served as CCN, many models abandon thereafter the pursuit of the aerosol mass in drops and ice. This might appear plausible when we assume that all drops and ice crystals end up as precipitation on the ground. But this is not true as drops evaporate and ice particles sublimate in the atmosphere. (This is the real setup and holds also for mixed phase and cold clouds). Why should we ignore the remaining aerosol mass after a cloud particle cycle? We also respect the water vapor budget when droplets evaporate!
The significance of the aerosol concentration is most obvious in this study where all precipitation characteristics are influenced by the different aerosol concentration which we applied.

© Pg. 4 line 5: Again, facing your goals the reader should understand how main features of the microphysical scheme works. Raindrops of 10mm are extremely rare (some thinks they just do not exist); as such it is important to understand how the scheme handles these potentially numerical artifacts of very large rain drops that aren't stable. This affects for sure your rain size distribution.
The two largest bins in our numerical grid for drops have no physical importance. They serve to guarantee the mass conservation when collision-coalescence is calculated. Once formed, they are immediately redistributed to smaller sizes by stochastic break-up (Hall, 1980). A complete presentation of the microphysical scheme is given in Flossmann and Wobrock (2010), referenced in the text.

© Pg. 6 line 18: Why do you start with describing Figure 4? (minor)
Figure 3a (now 4a) is used first - in the paragraph of the section 3. So we didn't change the order of the figures.

© Pg. 6 line 20: should be sixth moment, not "sixth momentum"? what do you mean in "normalized" here? (minor)

Done. The radar reflectivity $Z$ is normally in $mm^6/m^3$ but the radar reflectivity factor commonly used in order to facilitate comparisons between observations from different radars is in dBZ using:
$Z_{[in\ dBZ]} = 10\ log\ (Z_{[in\ mm6/m3]})$. That is, what we meant by 'normalized'. We clarified the text.

© Pg. 7 line 3: can you please explain from physical perspective what prevent the model (dynamical core + microphysical scheme) from being able to reproduce the change in orientation?

This is essentially due to the initial large scale data set. The model was initialized by the 3D fields of wind, temperature and humidity of ECMWF at 0:00 UTC with a horizontal resolution of about 44km in x and 55 km in y direction. All parameters are horizontally quite homogeneous, especially the humidity field over the Mediterranean Sea. The low level air masses responsible for the cloud formation in the morning were advected during night from the Balearic Islands. The coarse resolution of the outmost model (dx=8km) did not notably change the initial homogeneous structure of temperature and humidity and thus the inflowing air kept quite uniform until 10 UTC without a change in orientation. Thus, horizontal patches of air deviating in humidity, temperature or momentum from the large-scale conditions, did not developed in the simulations.

© Pg. 7 line 16-19: can you please comment from the microphysical scheme perspective -- why the area of rain accumulation is different, especially the area of the accumulated rain of ~38 (mm) and below is significantly underestimated. It looks like the scheme (or the setup) has problems in simulating shallow convection and/or stratiform clouds.

We attribute the missing rainfall over the western area of the mountains again to a lack in humidity and also to differences in wind with the real conditions. From ground radar observations (now Fig.5) and also from the profiles observed by the airborne cloud radar (not illustrated) we can exclude the presence of shallow convection or stratiform precipitation.

As a follow up query, how was the corresponding forecast of the 1M/2M/3M bulk microphysical schemes? Could you please comment on that.
The simulations available for IOP7a in the article of Hally et al (2014) don't give insights in the precipitation structure but restrict to 24 h rainfall amounts.

© Pg. 7 line 30: Indeed, but the reader may ask himself what in the microphysical scheme lead to this changes? If the paper would have a more coherent microphysical analysis (vertical cloud structure) you would be able to explain that from physical point of view.
Adding a microphysical analysis of the vertical cloud structure for the different aerosol scenarios comprises another research article, and thus we refrain for including it in this paper. In order to better communicate the vertical structure of cloud field, we already added the new Figs. 6a and b.

In order to moderate the Reviewer some deeper insights into the differences in cloud microphysics between the scenarios, we added Fig.R1 wherein two contoured frequency diagrams of the relative humidity with altitude are given, one for the Remote case, another for the HymRef. The frequency analysis uses 8300 soundings and restrict to the strongest area of the cloud and precipitation formation. Modeled relative humidity (RH) in the range from 90 to 103% was counted for the PDF, which was binned by 1%. We can see that in the Remote case that RH dominates between 100 to 101 % in the lower part from 1.3 to 5 km. In the levels from 5 to 6.7 km most RH take even 101 to 102%. For the HymRef scenario with higher aerosol loading, RH values between 100 - 101% contribute dominantly over all altitudes up to 6.7 km but the shift to 102% for z > 5 km is negligible. In addition, for altitudes above 4 km a significant part (30-35%) of the simulated RH remain below 100%.
This result clearly demonstrates that the cleaner atmosphere allows the formation of higher supersaturation which consequently effects a stronger phase transition to water and ice, explaining the increase in surface rain for the Remote case.

In addition, this is an example to the systematic problem in the ASD setup, where the Remote setup has 600 #/cm3 and 250 #/cm3 in the accumulation and the coarse modes, respectively. These modes are

readily nucleate to droplets in any typical deep convection systems, and should lead to early rain formation (especially the coarse mode with 250 #/cm3). This is quite different aerosols regime.

As already explained above, there is a typo for the coarse mode diameters in the Remote case in Table 1. This is now corrected.

© Pg. 8 lines 10-11: Have you checked your low-surface "cold pool"? The question is whether the limited spatial changes in rain results from a dynamical reason or underestimation in low-level rain amount and sizes which limits the evaporation and thus decrease the "cold pool". This is related to the large scale forcing vs. local convective instability.

Cold pools do not develop over the quite complex terrain and under high humidity conditions as for this as given for this event. The contribution of evaporation of rain is negligible as the cloud base is low and the subjacent air still holds a high relative humidity (see new Fig. 6).

Cold pools as a consequence of heavy rain in the southern part of France do typically from over the Mediterranean Sea as demonstrated by Duffourg et al (2016) or Martinet et al (2017).

© Pg. 9 lines 4-5: Regarding your conclusion that "more rain occurs when low particle number prevails" – this is likely to be true for 2 ASDs with different number concentration per modal size in a warm convective system. When you convolve number concentration between ASD modes, the rain can be initiated from different level in the cloud.

As explained above, there was a mistake in the mean diameters for modes 2 and 3 for the Remote aerosol spectrum in Table 1.

As the convection becomes deep enough, lower CCN size penetrates to areas where high vertical velocity occurs and thus higher supersaturation above liquid/ice occurs (Sw, Si), and more smaller drops nucleates, which means more vapor is extracted from the atmospheric column (Si > Sw) compared to nucleation at lower levels where Sw is limited by warm rain; this serves as positive feedback that intensify the convection as more drop freezes at higher levels, as well as lead to increase in large/dense hydrometeor size which sediment and force downdraft and further positive feedback.

It is right that high velocity occurs and higher supersaturation forms in 5-7 km as illustrated in the CFAD Fig.R1 for RH in the *Remote* scenario. But the higher supersaturation is due the lack of new droplet activation from aerosols. The number and the size of non-activated CCN are low in these cloud levels. Even if they nucleate their number contribution and mass is negligible compared to the already existing hydrometeor number and mass. In addition, diffusivity of water vapor and thermal conductivity are strongly reduced for temperatures below -15° and the supersaturation peaks needed for nucleation have to hold for quite long periods to allow activation of small particles. Fig.R2 (from Leroy et al, 2007, Atm. Research) gives the time scales for activation a particle with D= 40 nm under different conditions of supersaturation and temperature. In this temperature range from -15 to -25 °C we have to calculate aerosol activation time dependent and cannot apply equilibrium Köhler theory.

The above is called 'convective invigoration' that leads to more intense rain rate. In your Remote ASD setup you are not only reducing total aerosol number concentration, but you also "pushes" the clouds to rain-out (warm-rain) substantially earlier due to the increased number concentration in the accumulation and especially the coarse modes. Therefore, based on the limited analysis presented here, your conclusion needs to be rephrased to include the information about the differences in ASDs modal concentration

Number concentration in the accumulation and especially the coarse modes are not increased. Again sorry for our mistake in Table 1

© Pg. 9 lines 12-13: again, you have forced at least 2 more degrees of freedom in that the Remote ASD has substantially different aerosol number concentration distributed between the modes. You need to address this by dedicated sensitivity test, or at least restrict your conclusions.

See above.

© Pg. 10 line 24: the value of 9mm / 5min in Figure 9 cannot be seen. Please comment or correct this value. (minor)

We modified the phrase on page 10 line 24.

© Pg. 11 lines 25-30: There is no indication of the temperature near the surface and where is the freezing level placed.

This is done now by adding the new Fig.6b)

There is no clear indication how the averaging has been performed (space-wise). Since the model simulations clearly underestimates the area of ~25 mm (and below) and the averaging was made between 900-1000m, it is not clear to me how the RWC = 0.5 g/m3 rain size distribution in Figure 11b are reasonably compared to observations (as shown in Figure 11a).

All model grid point at z=0 m and an underlying topography between 900 to1000 m were selected when their RWC was in between 0.4-0.6 g/m$^3$.

Such RWC are largely in the underestimated area and can be attributed to shallow convection or even to heavily stratiform precipitating clouds. Can you explain this apparent discrepancy?

The Radar observations (as well as the model results) don't give any hint for shallow convection or even to heavily stratiform precipitating clouds.

Furthermore, in principle raindrops grows at the expense of small-medium size raindrops (0.3 – 1.5 mm) as these fall through the cloudy area, but this is quite a simplistic point of view as observations (Figure 11a) indicates that other processes are likely to be responsible for the ongoing supply of these small-medium raindrop near the surface and for vast range of RWC. These processes are being determined well above the surface (for instance: melting process; breakup of large raindrop). Thus, based on this simplistic microphysical analysis made here, the conclusion drawn should be very careful as probably the model has some drawbacks in this aspect.

You are right. We learned from this model study that precipitation microphysics can be improved in DESCAM and another article will focus on this subject.

Pg. 12 line 22: what is the context for "superficial" here? (minor)

Was changed

**Technical comments**

© Pg. 2 line 27: There is no meaning in "bin resolved"; you probably mean "size resolved".

The term "Bin-resolved" is extensively used in the literature, as bulk schemes also give size resolved information, in a parameterized way

© Pg. 3 line 26: Maybe "outermost model domain" is preferable. Also, the resolution increases, where the grid spacing decreases.

Done

© Pg. 4 line 1: A microphysical scheme (like the DESCAM) calculates (or prognoses) the temporal and spatial changes in the distribution functions. The overall set up of the dynamical core coupled to the microphysical scheme with the BC/IC "simulates" a particular test case and the corresponding fields (rain, CWC, RWC, etc.).

Done

© Pg. 4 line 26: … a third aerosol distribution with lower number concentration is used.

Done

© Pg. 4 line 33 (and throughout the text): number distribution is confusing. Use number concentration or/and aerosol/droplet/rain size distribution.

Done

© Pg. 11 lines 3: rain size distribution should be noted as RSD and not DSD. DSD is droplet/drops size distribution.

Done

[Figure]

*Fig.R1: Contoured frequency diagram of relative humidity RH as a function of altitude. The color scale give the frequency from 0 to 77%.*

*Relative humidity is given in bins of 1% from 90 to 103%. Altitudes a.s.l range from 750 to 9750 m with an increment of 150 m.*

[Figure]

**a)**

Fig. R2: Time of activation of a D=40 nm aerosol particle from its equilibrium size at 99% to 1 µm droplet size as a function of temperature and supersaturation.

---

## Editor Decision (ED1)

Second review of Kagkara et al. 2020

Thanks to the authors for the detailed answers. With the correction on aerosol populations, the paper now effectively demonstrates both that the bin scheme DESCAM is able to produce reasonable amounts of precipitation in a 3D, real-case simulation of a convective system, and that there is a sensitivity to the aerosol population, therefore pushing for the use of aerosol-aware cloud physics schemes. My remaining comments are :

General comments

- The vertical cross-sections are a welcome addition which provide a better view of the simulated clouds, but, I expected a bit more in terms of processes leading to the ground level rain characteristics. The presented results are interesting, and the authors state that the focus on rainfall simulation fits the NHESS journal well, but as a cloud physics scientist, I still miss some physical process understanding, which are briefly mentioned in the conclusion, such as :
      - how is the rain size distribution evolving with height and is this evolution depending on the number of aerosol (even if we have no observations to compare to) ?
      why are the lower precipitation amounts underestimated, is this only due to initial & coupling conditions or also linked to microphysics or other processes (turbulent mixing, entrainment, dynamics,...) and is this a usual feature of specific to this case ?

- The correction on aerosol populations answers the main issue with the paper, as the new Figure 2 shows that the three aerosol populations are in fact ordered from the high CCN concentrations (HymRef) to the low CCN concentration (Remote), (almost) consistently for all particle diameters. This still seems cumbersome (it would have been easier to, e.g., divide the real population concentration by 2 and 5, and keep the same size distribution shape), but there is no issue with that anymore. Regarding aerosols, I still have other questions :
      - Above 3km, the concentration is fixed at ~900/cm3, so the same value for all cases, so the studied aerosol impact is only linked to the aerosols at cloud base, and those transported inside the cloud by updrafts, and neglects the effect of aerosol entrainment from cloud sides/top during the cloud formation. This is stated in the authors' answers, is not a problem but should be mentioned in the manuscript.
      - Maybe the new Fig.2 could also include a second panel showing the aerosol number concentration (sum of the three modes) for each experiment along the vertical ?

minor comments
- p2 l15 : Tauffour et al. (…) with a the two-moment scheme (…)
- p2 l29-30 : Although most studies using bin schemes are performed in 2D or idealized configurations, some bin schemes have already successfully been used for real cases of deep convection, (although not for HyMeX cases), even for aerosol-cloud interactions assessment (eg. Iguchi et al 2008, Fan et al. 2012). So, here and in the conclusion, maybe this could be modified : "test if the DESCAM bin scheme is able to ..." ?
- p2 l31: Although bulk models are indeed less precise than bin schemes, they usually perform well enough for convection and are able to produce high amounts of precipitation. Studies of HyMeX cases cited in this paper indeed prove that point (Hally 2014, Duffourg 2016, etc), especially for cases involving strong synoptic forcing of orographic lifting. Although some errors and/or uncertainty remain, they are not attributable to the microphysics only. The same can be said for this case using the DESCAM bin scheme (indeed, the conclusion states that some differences with observations may very well be due to the initial and lateral boundary conditions). "Often have

difficulties" is a bit overstated and mixes all uncertainty sources in simulations. Maybe change for something like "rely on much more assumptions and approximations to predict ..." ?
- p13 l14 : see comment above about other bin schemes used in 3D real case simulations
- p15 l.33 : See comment above about bulk schemes. The statement "better represented as they are generally in bulk models" is vague and not justified. Again, of course they can be improved and the bin scheme is valuable in this regard, but bulk schemes have been used successfully for high impact weather forecasts and warnings for quite some time, and generally produce reasonable amounts of rain for Mediterranean heavy precipitating cases.

Ref :

Fan, J., L. R. Leung, Z. Li, H. Morrison, H. Chen, Y. Zhou, Y. Qian, and Y. Wang (2012), Aerosol impacts on clouds and precipitation in eastern China: Results from bin and bulk microphysics, *J. Geophys. Res. Atmos.*, *117*(D16), n/a–n/a, doi:10.1029/2011JD016537.

Iguchi, T., Nakajima, T., Khain, A. P., Saito, K., Takemura, T., and Suzuki, K. ( 2008), Modeling the influence of aerosols on cloud microphysical properties in the east Asia region using a mesoscale model coupled with a bin-based cloud microphysics scheme, *J. Geophys. Res.*, 113, D14215, doi:10.1029/2007JD009774.

---

## Author Response (AR2)

**NHESS-2019-321**

**Responses to reviews**

The sensitivity of intense rainfall to aerosol particle loading - a comparison of bin-resolved microphysics modelling with observations of heavy precipitation from HyMeX IOP7a

C. Kagkara, W. Wobrock, C. Planche and A. I. Flossmann

25 March, 2020

**Note to the editor**

We thank all of the reviewers, whose comments have led to significant improvements in the analysis and, thus, our manuscript. We are grateful that reviewer#1 (and #2) found that the manuscript is much improved, and close to ready for publication.

Changes in the manuscript and the reply to the individual remarks of reviewer #2 are marked in red for easier notice.

**Responses to reviewer #1's comments**

I've read the authors response, and most of the points were addressed reasonably well. The manuscript seems appropriate to the NHESS journal and I do not have any particular objection or further inquiry for/before acceptance of the manuscript.
Thanks.

**Responses to reviewer #2's comments**

Thanks to the authors for the detailed answers. With the correction on aerosol populations, the paper now effectively demonstrates both that the bin scheme DESCAM is able to produce reasonable amounts of precipitation in a 3D, real-case simulation of a convective system, and that there is a sensitivity to the aerosol population, therefore pushing for the use of aerosol-aware cloud physics schemes.
My remaining comments are:

**General comments**

- The vertical cross-sections are a welcome addition which provide a better view of the simulated clouds, but, I expected a bit more in terms of processes leading to the ground level rain characteristics. The presented results are interesting, and the authors state that the focus on rainfall simulation fits the NHESS journal well, but as a cloud physics scientist, I still miss some physical process understanding, which are briefly mentioned in the conclusion, such as:
  - how is the rain size distribution evolving with height ? and is this evolution depending on the number of aerosol (even if we have no observations to compare to) ?

We selected for the same model time as in Fig. 6 (8:20 UTC, see the revised paper) three cloud-layers (at 1150, 2150 and 3150 m above sea level) each with a depth of 300 m (i.e. ±150 m). For each layer the modeled droplets spectra were averaged amounting typically to 2000 spectra per layer. Figure R3a shows the resulting number distribution, Fig R3b the mass size distribution. The continuous lines give the spectra of *HymRef* for the three layers. With decreasing altitude, the raindrop size distributions illustrate that the number of small drop sizes decreases while the number of larger sizes (> 2mm) increases. The increase of the large drop sizes with decreasing altitude becomes more obvious in the illustration for the mass distributions of Fig. R3b. This behavior confirms our statement, that collection-coalescence is responsible for the shift of the water mass to the larger raindrops.

[Figure]

[Figure]

Fig. R3: Modeled number distributions (a and c in $m^{-3}mm^{-1}$) and mass distributions (b in $g\ m^{-3}\ mm^{-1}$) for the three atmospheric layers below the melting level at 8:20 UTC. The grey area in c highlights the size interval represented in a.

The modeled spectra of the *Remote* case at 1150 and 3150 m were also depicted in Figs R3a and b (dashed lines). This allows a comparison of *HymRef* spectra with those of clean atmospheric conditions. We note, in agreement with our findings detailed in the paper, that a lower aerosol number in the initial atmospheric conditions leads to larger raindrop sizes. This analysis also confirms this vertical

behavior over the entire layer where warm rain dominates. This result confirms the findings already explained in the paper, we did not include these additional Figs in the paper.

When comparing the droplet numbers between *Remote* and *HymRef* at 3150 m, it is surprising to see that also small raindrops in the diameter range from 0.1 to 0.7 mm are more frequent in the *Remote* scenario. In order to understand what this result of *HymRef*, we extended the drop size distributions to size ranges of cloud droplets (down to 10µm). Fig. R3c shows that the high concentration of small drops formed in *HymRef* restricts to sizes below 30 µm. This again is coherent with our statements on the effect of aerosol number concentration on the cloud droplet evolution.

Differences between the mean spectra, especially for raindrop sizes > 2 mm, are quite weak (see Figs R3a and b). Results for the mean spectra depend strongly on the horizontal location of the selected grid points. As noted above, altitudes were taken above sea level. Due to the complex terrain of the Cevennes and Vivarais Mountains, the vertical distance between underlying topography and e.g. 3150 m varies between 1850 to 2850 m. Thus, cloud modeling over complex terrain makes it quite difficult to distinguish and to explain the processes dominating for cloud and precipitation formation. The permanent changes in up- and downdrafts modify continuously the field of relative humidity causing regions with strong condensation rates and others with evaporation and strong rainfall. Thus, detailed physical process understanding has to be locally restricted to regions where dynamical, thermodynamical and thus microphysical conditions are similar.

o  why are the lower precipitation amounts underestimated, is this only due to initial & coupling conditions or also linked to microphysics or other processes (turbulent mixing, entrainment, dynamics,...) and is this a usual feature of specific to this case ?

This is definitely due to the initial conditions. We run the same case with WRF for an identical model setup (initial and boundary conditions, size and resolution of outermost and nested domains). WRF (using the Morrison or the Thompson scheme) produces the same location and horizontal extension for surface rain over the Cevennes Mountains.

Unfortunately we cannot add here (i.e. on the public site of Copernicus) a figure of this comparison with WRF, as it is part of a paper that will be submitted to another journal.

• The correction on aerosol populations answers the main issue with the paper, as the new Figure 2 shows that the three aerosol populations are in fact ordered from the high CCN concentrations (*HymRef*) to the low CCN concentration (*Remote*), (almost) consistently for all particle diameters. This still seems cumbersome (it would have been easier to, e.g., divide the real population concentration by 2 and 5, and keep the same size distribution shape), but there is no issue with that anymore. Regarding aerosols, I still have other questions:

o  above 3km, the concentration is fixed at ~900/cm3, so the same value for all cases, so the studied aerosol impact is only linked to the aerosols at cloud base, and those transported inside the cloud by updrafts, and neglects the effect of aerosol entrainment from cloud sides/top during the cloud formation. This is stated in the authors' answers, is not a problem but should be mentioned in the manuscript.

Initial number concentrations of aerosols above 3 km differ for each scenario. We join this information, as proposed by the reviewer, in the new Fig 2b. Our simulations use a 3D Eulerian model,

wherein all prognostic variables (i.e. also each aerosol, drop and particle bin) are transported by advection, sedimentation and turbulent mixing in all possible directions. Thus, no effect of aerosol entrainment from cloud sides/top is neglected.

- o Maybe the new Fig.2 could also include a second panel showing the aerosol number concentration (sum of the three modes) for each experiment along the vertical?
  done

**Minor comments**

- p2 l15 : Tauour et al. (…) with  **the** two-moment scheme (…)
  Corrected

- p2 l29-30 : Although most studies using bin schemes are performed in 2D or idealized configurations, some bin schemes have already successfully been used for real cases of deep convection, (although not for HyMeX cases), even for aerosol-cloud interactions assessment (eg. Iguchi et al 2008, Fan et al. 2012). So, here and in the conclusion, maybe this could be modified: "test if the DESCAM bin scheme is able to ..." ?
  We modified the introduction and the conclusion as followed:

  - in the introduction section, before the description of the main objective of the paper, we included the following text:
  "*Only few studies (e.g. Iguchi et al., 2008; Fan et al., 2012) have been focused on real deep convective systems with a bin microphysics scheme in a 3D dynamical framework, and none of them was applied to an intense precipitating system as usually observed in autumn over the western Mediterranean basin.*"

  - the conclusion was modified as follow:
  "*A major objective of this study was to test if a bin resolved microphysics module in a 3D mesoscale model is successful in reproducing a real case of intense precipitation usually observed over the western Mediterranean basin.*"

- p2 l31: Although bulk models are indeed less precise than bin schemes, they usually perform well enough for convection and are able to produce high amounts of precipitation. Studies of HyMeX cases cited in this paper indeed prove that point (Hally 2014, Duffourg 2016, etc), especially for cases involving strong synoptic forcing of orographic lifting. Although some errors and/or uncertainty remain, they are not attributable to the microphysics only. The same can be said for this case using the DESCAM bin scheme (indeed, the conclusion states that some differences with observations may very well be due to the initial and lateral boundary conditions). "Often have difficulties" is a bit overstated and mixes all uncertainty sources in simulations. Maybe change for something like "rely on much more assumptions and approximations to predict ..." ?

We modified following your suggestion.

- p13 l.14: see comment above about other bin schemes used in 3D real case simulations
  See our response above.

- p15 l.33: See comment above about bulk schemes. The statement "better represented as they are generally in bulk models" is vague and not justified. Again, of course they can be improved and the bin scheme is valuable in this regard, but bulk schemes have been used successfully for high impact weather forecasts and warnings for quite some time, and generally produce reasonable amounts of rain for Mediterranean heavy precipitating cases.
  We clarified the manuscript: "*Regarding the other objective of the current investigation, our study showed the potential of a bin-resolved modelling to reproduce the heavy precipitation periods usually observed over the Cevennes area. Even though the weaker precipitation was underestimated in the model, the peak values that would warrant an alert to the population were well represented. This bin-resolved modelling also provides a better understanding of the rain microphysics processes compared to bulk models as the microphysics is explicitly represented.*"

*References:*

*Fan, J., L. R. Leung, Z. Li, H. Morrison, H. Chen, Y. Zhou, Y. Qian, and Y. Wang (2012), Aerosol impacts on clouds and precipitation in eastern China: Results from bin and bulk microphysics, J. Geophys. Res. Atmos., 117(D16), n/a–n/a, doi:10.1029/2011JD016537.*

*Iguchi, T., Nakajima, T., Khain, A. P., Saito, K., Takemura, T., and Suzuki, K. (2008), Modeling the influence of aerosols on cloud microphysical properties in the east Asia region using a mesoscale model coupled with a bin -based cloud microphysics scheme, based cloud microphysics scheme, J. Geophys. Res., 113, D14215, doi:10.1029/2007JD009774.*

---

## Author Response (AR3)

Dear Editor,
we submitted the corrected version of the manuscript. It takes into account your requests (below), except for some model names where a reference is given next to them in the text.
We hope that you find the manuscript now in an acceptable form.

We like to thank you for your effort.

kind regards
Wolfram Wobrock
* * *
**Editor Decision: Publish subject to technical corrections** (08 Apr 2020) by Christian Barthlott
Comments to the Author:
Dear authors,

thank you for revising your manuscript and your responses to the review comments. All referee comments have been addressed satisfactorily, therefore I am pleased to accept your study for publication. Please make these technical corrections before uploading the files to the production office:

p2 l11: explain IFS/ECMWF and other model abbreviations
p9 l26: have already be done --> been done
p10 l18: interval --> intervals
p13, l25 homogenous --> homogeneous

Please also avoid paragraphs with just one (or two) sentence(s). For example, the first 3 paragraphs of the introduction could be merged into one single paragraph. This can be done at various places in the manuscript.

Best wishes,

Christian Barthlott